# CLAUSE: Agentic Neuro-Symbolic Knowledge Graph Reasoning via Dynamic Learnable Context Engineering

**Yang Zhao**[1]*, **Chengxiao Dai**[2]*, **Wei Zhuo**[3], **Yue Xiu**[1], **Dusit Niyato**[3]

[1]Independent Researcher.
[2]School of Computer Science, The University of Sydney, Australia.
[3]College of Computing & Data Science, Nanyang Technological University, Singapore.

## Abstract

Knowledge graphs provide structured context for multi-hop question answering, but deployed systems must balance answer accuracy with strict latency and cost targets while preserving provenance. Static $k$-hop expansions and "think-longer" prompting often over-retrieve, inflate context, and yield unpredictable runtime. Thus, we introduce CLAUSE, an agentic three-agent neuro-symbolic framework that treats context construction as a sequential decision process over knowledge graphs, deciding what to expand, which paths to follow or backtrack, what evidence to keep and when to stop. Latency (interaction steps) and prompt cost (selected tokens) are exposed as user-specified budgets or prices, allowing per-query adaptation to trade-offs among accuracy, latency, and cost without retraining. CLAUSE employs the proposed Lagrangian-Constrained Multi-Agent Proximal Policy Optimization (LC-MAPPO) algorithm to coordinate three agents: *Subgraph Architect*, *Path Navigator*, and *Context Curator*, so that subgraph construction, reasoning path discovery, and evidence selection are jointly optimized under per-query resource budgets on edge edits, interaction steps, and selected tokens. Across HotpotQA, MetaQA, and FactKG, CLAUSE yields higher EM@1 while reducing subgraph growth and end-to-end latency at equal or lower token budgets. On MetaQA-2-hop, relative to the strongest RAG baseline (GraphRAG), CLAUSE achieves +39.3 EM@1 with 18.6% lower latency, and 40.9% lower edge growth. The resulting contexts are compact, provenance-preserving, and deliver predictable performance under deployment constraints.

## 1 Introduction

Large language models (LLMs) benefit from external structure for knowledge graph question answering (KGQA) that require multi-hop reasoning and provenance (Lewis et al., 2020; Yang et al., 2018; Pan et al., 2023). Knowledge graphs (KGs) are a natural substrate: they expose typed entities and relations, support symbolic traversals, and yield auditable context trails (Yasunaga et al., 2021; Das et al., 2018). A common design is to build a query-based local neighborhood in the KG, and then condition a reader language model to produce the answer (Sun et al., 2019; Ding et al., 2024).

How the graph context is assembled often misaligns with both answer quality and runtime constraints. Fixed $k$-hop expansions serialize many triples, inflating token mass and latency (Zhou et al., 2024; Wan et al., 2024) and introducing distractors that depress accuracy (Jiang & Bansal, 2019). Extending chain-of-thought (Wei et al., 2022; Kojima et al., 2022) lengthens per-step reasoning without changing *which* evidence is visible and offers little control over end-to-end latency (Zhou et al., 2024). In practice, systems are constrained not only by prompt length but also by the number of interaction steps, how often we edit, traverse, and curate, yet most pipelines expose only heuristic knobs (hop depth, degree caps, top-$k$).

Our view is to make context construction itself as the learning problem: decide which edges to add or delete, which paths to pursue or backtrack, which snippets to keep, and when to stop, all under explicit

---

*Equal contribution. Corresponding author: Yang Zhao (`s180049@e.ntu.edu.sg`).

caps or prices on interaction steps and selected tokens. This replaces brittle $k$-hop heuristics with a learned, budget-aware controller and makes accuracy–latency–cost trade-offs explicit and tunable. We then propose CLAUSE, an *agentic neuro-symbolic* framework with three agents—*Subgraph Architect*, *Path Navigator*, and *Context Curator*. Decisions unfold sequentially on a symbolic state (nodes, edges, paths) through discrete, auditable actions (edit, traverse, curate), while compact neural scorers prioritize entities, relations, and neighborhoods. In this design, step and token usage enter the training objective directly, so stopping rules and exploration depth are learned rather than hard-coded.

Specifically, three cooperative agents operate on the KG: *Subgraph Architect* constructs a question-anchored subgraph that preserves answer-supporting paths while avoiding over-expansion; the *Path Navigator* discovers and revises reasoning paths while respecting a step budget; and *Context Curator* assembles a minimal set of textualized snippets sufficient for accurate responses from LLMs under a token budget. We coordinate three agents with LC-MAPPO—a Lagrangian-constrained centralized training with decentralized execution (CTDE) variant of PPO that uses a centralized critic and Lagrangian dual variables to learn decentralized policies, which maximize task reward while enforcing per-query budgets on edge edits, interaction steps, and selected tokens (Foerster et al., 2018; Rashid et al., 2020; Schulman et al., 2017; Yu et al., 2022; Achiam et al., 2017; Stooke et al., 2020). During inference, a single checkpoint runs under hard budgets (caps) or fixed prices (soft trade-offs), adapting per query without retraining.

Empirically, the CLAUSE framework produces compact, targeted contexts and predictable runtime. In HotpotQA, MetaQA, and FactKG, it reduces edge counts and end-to-end latency while improving exact match at the matched token mass, as shown in Section 5. Requirement sweeps reveal clear accuracy–latency–cost Pareto frontiers: shifting budget from per-step reasoning to interaction improves accuracy at fixed tokens, and tightening the step budget reduces latency with little or no loss in accuracy.

**Contributions.** (1) *Formulation.* We cast multi-hop KGQA as *requirements-conditioned* context assembly with per-query budgets/prices on three deployment-relevant resources: (i) subgraph edits, (ii) interaction steps (latency proxy), and (iii) selected tokens (prompt cost). This makes accuracy–efficiency trade-offs explicit and tunable.
(2) *Framework.* CLAUSE is an agentic neuro-symbolic controller that *jointly* decides what to edit, which paths to follow or backtrack, what textual evidence to keep, and when to STOP. Actions are symbolic (auditable) and priorities come from lightweight neural scorers, yielding compact, provenance-preserving context.
(3) *Training.* We adapt constrained RL to this setting via LC-MAPPO: centralized training with decentralized execution, a multi-head critic that separates *task* value from *edge/step/token* costs, and per-budget dual variables that enforce episode-level requirements or enable price-based trade-offs at test time.
(4) *Evidence.* In HotpotQA, MetaQA, and FactKG, CLAUSE achieves higher or matched EM at equal or lower budgets, with reduced subgraph growth and latency. Ablations show that removing any agent or constraint handling degrades accuracy and/or efficiency, supporting the need for joint control and explicit budgets.

## 2 PRELIMINARIES AND RELATED WORK

### 2.1 PRELIMINARIES

**Neuro-symbolic definition.** We view *neuro-symbolic inference* as coupling an explicit symbolic calculus (Boolean, first-order, or soft/fuzzy) with a learned scoring/belief module; differentiable logic is unnecessary—only a principled linkage between symbols and learned scores is required (Smet & Raedt, 2025). **KGQA as neuro-symbolic.** KGQA operates on typed entity–relation graphs and commonly targets (i) single-relation queries, (ii) multi-hop path queries, and (iii) compositional-logic queries (e.g., conjunction/disjunction/negation) (Zhang et al., 2021). Surveys group approaches into (1) logic-informed embeddings, (2) embeddings trained with logical constraints, and (3) *rule/path learning* where a neural controller searches over symbolic paths/rules (DeLong et al., 2025). We adopt (3): a dynamic learnable agentic framework edits a KG for reasoning.

## 2.2 RELATED WORK

**Existing Multi-hop KGQA Solutions.** Multi-hop KGQA must balance accuracy and provenance with strict constraints on latency and prompt cost. In practice, two resources dominate deployment behavior: the number of *interaction steps* taken while assembling context and the *selected tokens* ultimately shown to the reader LLM. Static $k$-hop expansions often over-retrieve, inflate prompts, and surface distractors (Zhou et al., 2024; Wan et al., 2024; Jiang & Bansal, 2019), while typical pipelines expose only heuristic knobs rather than learned, per-query control. A long line of *symbolic/neuro-symbolic* KGQA operates directly on entity–relation structure. Path-following and rule-learning systems (e.g., MINERVA, NeuralLP, TensorLog, RNNLogic) traverse the graph to derive answers (Das et al., 2018; Yang et al., 2017; Cohen, 2016; Qu et al., 2021); graph-aware readers (e.g., QAGNN) inject KG signals into the encoder (Yasunaga et al., 2021). Question-conditioned subgraph builders such as GraftNet and PullNet assemble local neighborhoods for a downstream reader (Sun et al., 2019). These approaches typically set expansion depth/degree and filtering thresholds a priori, which makes runtime behavior sensitive to manual tuning and obscures the accuracy–efficiency trade-off. Then, work on *context engineering* shows that prompt composition strongly affects both cost and accuracy (Zhou et al., 2024; Wan et al., 2024). Chain-of-thought prompting can help certain tasks (Wei et al., 2022; Kojima et al., 2022; Han et al., 2025), yet it primarily lengthens the reasoning text without changing *which* evidence is visible, offering limited leverage over end-to-end latency (Zhou et al., 2024). Moreover, *Retrieval-augmented generation* (RAG) conditions generation on external evidence (Lewis et al., 2020; Karpukhin et al., 2020; Izacard et al., 2023). Recent variants interleave reasoning and retrieval (ReAct) (Yao et al., 2023), incorporate self-feedback (SELF-RAG) (Asai et al., 2024), or adapt retrieval frequency to difficulty/confidence (Jeong et al., 2024; Zhang et al., 2024). Graph-guided pipelines (e.g., GraphRAG; Think-on-Graph 2.0) leverage entity–relation structure for multi-hop collection (Edge et al., 2025; Ma et al., 2025). These systems often rely on fixed hop limits or hand-tuned schedules; optimization of construction, traversal, and selection is rarely carried out jointly under explicit step/token costs. Finally, *agentic LLMs* plan, call tools, and decide when to act versus reflect (Press et al., 2023; Yao et al., 2023; Shinn et al., 2023; Schick et al., 2023; Shen et al., 2023; Liu et al., 2024; Wang et al., 2023). Their flexibility comes with multi-step deliberation that can raise interaction cost, and per-episode resource control is implicit.

**MARL and constrained optimization.** Multi-agent reinforcement learning (MARL) addresses decentralized coordination under partial observability and non-stationarity, where multiple local-view actors must produce joint behavior that optimizes a global objective subject to deployment constraints (e.g., latency, token budget, graph edits). A widely used recipe is centralized training with decentralized execution (CTDE), which stabilizes learning and credit assignment via a centralized value while keeping actors decentralized at test time; representative instances include COMA, which introduces a counterfactual baseline for per-agent credit, and QMIX, which learns a monotonic mixing network to factorize joint values into per-agent utilities (Foerster et al., 2018; Rashid et al., 2020). Building on PPO (Schulman et al., 2017), MAPPO shows that PPO-style updates with a centralized critic are strong, simple baselines on standard cooperative benchmarks (Yu et al., 2022). Existing methods largely fall into two families: value factorization (e.g., QMIX), which is efficient and scalable but restricted by the monotonic mixing constraint and can misattribute credit when joint action values are non-monotonic; and policy-gradient CTDE (e.g., COMA/MAPPO), which is flexible but higher-variance/sample-hungry and, in vanilla form, lacks principled mechanisms to enforce per-episode resource constraints. Single-penalty constrained RL such as Reward-Constrained Policy Optimization (RCPO) (Tessler et al., 2019) further conflates heterogeneous costs, making it difficult to independently control edge growth, interaction steps, and selected tokens. Preference-optimization methods (GRPO/DPO) instead learn from static pairwise/group preferences over complete responses in bandit-like, text-only settings without explicit environment state transitions (Rafailov et al., 2023; Shao et al., 2024); they neither decompose multi-agent credit nor estimate shaped values on graph states, and they provide no handle for enforcing per-episode constraints.

**Positioning.** In summary, we situate CLAUSE among four families: (i) *question-conditioned subgraph builders* and graph-guided RAG (e.g., GraftNet, PullNet, GraphRAG; (Sun et al., 2019; Edge et al., 2025)), which rely on fixed hop/degree/top-$k$ rules; (ii) *path/rule learners* (MINERVA, NeuralLP, RNNLogic; (Das et al., 2018; Yang et al., 2017; Qu et al., 2021)) that optimize task reward without explicit latency/token control; (iii) *agentic LLMs* (ReAct, Graph-of-Thoughts, AutoGen; (Yao et al., 2023; Besta et al., 2024; Wu et al., 2023)) that interleave tools but do not enforce per-episode resources; and (iv) *constrained RL* (RCPO or fixed-penalty PPO; MAPPO/COMA without

Table 1: Notation. See Appendix B for the extended table.

| Symbol | Description |
|---|---|
| $\mathcal{K} = (V, R, E)$ | Global knowledge graph; $E \subseteq V \times R \times V$ |
| $G_t = (V_t, E_t)$; $G^\star$ | Evolving subgraph at step $t$; final subgraph |
| $\mathcal{F}_t$; $\mathcal{P}_t$; $\mathcal{A}_t$ | Frontier nodes; candidate pool at step $t$; typed outgoing candidates (navigator actions) |
| $q$; $y$, $\hat{y}$ | Input question; gold / predicted answers |
| $p_t$; $\Pi$ | Path prefix at step $t$; set of discovered paths (provenance) |
| $\pi_B, \pi_T, \pi_S$; $a_t^B, a_t^T, a_t^S$ | Policies and actions for EDIT/TRAVERSE/CURATE |
| $s_t = (q, G_t, \mathcal{F}_t, \mathcal{P}_t, \mathbf{b}_t)$ | State summary at step $t$; $\mathbf{b}_t$: remaining budgets (vector) |
| $\beta = (\beta_{\text{edge}}, \beta_{\text{lat}}, \beta_{\text{tok}})$ | Episode budgets: edges, latency (steps), selected tokens |
| $C = (C_{\text{edge}}, C_{\text{lat}}, C_{\text{tok}})$; $c_t^{(k)}$ | Cumulative costs and per-step increments ($k \in \{\text{edge}, \text{lat}, \text{tok}\}$; $\sum_t c_t^{(k)} = C_k$) |
| $\boldsymbol{\lambda} = (\lambda_{\text{edge}}, \lambda_{\text{lat}}, \lambda_{\text{tok}})$ | Lagrange multipliers (resource prices) |
| $R_{\text{acc}}(\tau)$; $r_t^{\text{acc}}$; $r_t'$ | Episode reward; per-step task reward; shaped return $r_t' = r_t^{\text{acc}} - \sum_k \lambda_k c_t^{(k)}$ |
| $\mathcal{L}(\pi, \boldsymbol{\lambda})$ | Lagrangian $\mathbb{E}[R_{\text{acc}}(\tau) - \boldsymbol{\lambda}^\top C(\tau)]$ |
| $Q^{\text{task}}, Q^{\text{edge}}, Q^{\text{lat}}, Q^{\text{tok}}$ | Centralized-critic action-values (task head + three cost heads) |
| $A_t^{i,h}$, $A_t^{i,\lambda}$; $i \in \{B, T, S\}$ | Counterfactual and Lagrangian-shaped advantages; agent index |
| $D^\star$; $\text{tok}(\cdot)$ | Ordered curated evidence; token-count operator |
| $e = (u, r, v)$ | KG triple (head $u$, relation $r$, tail $v$) |
| $H$, $\bar{d}$, $|\mathcal{P}_t|$, $K$ | Hop cap; avg local branching factor; pool size; selected list length ($K$ dynamic) |

constraints; (Tessler et al., 2019; Schulman et al., 2017; Yu et al., 2022; Foerster et al., 2018)). In contrast, we cast KGQA as a *constrained* decision process with three deployment-relevant costs (edges/steps/tokens), learn *price-aware* edit/traverse/curate policies with explicit STOP, and train with *separate* cost heads and dual variables so a single checkpoint supports both budget caps and price trade-offs.

## 3 PROBLEM FORMULATION

**Problem.** We study multi-hop KGQA over a typed knowledge graph $\mathcal{K} = (V, R, E)$ with entities $V$, relation types $R$, and triples $E \subseteq V \times R \times V$. A query $q$ is natural language; the gold answer $y^\star \subseteq V$ can be a single entity or a set (surface strings, when provided, are canonicalized to IDs in $V$). Given $(\mathcal{K}, q)$, the system outputs a prediction $\hat{y} \subseteq V$ and a compact, provenance-preserving context for the reader LLM. An *episode* corresponds to one question.

**State and observations.** At round $t$, the controller maintains a working subgraph

$$G_t = (V_t, E_t), \qquad E_t \subseteq V_t \times R \times V_t,$$

a *frontier* $\mathcal{F}_t \subseteq V_t$ of nodes eligible for expansion, a candidate pool $\mathcal{P}_t$ of textualized units (nodes/edges/paths and optional retrieval hits), and remaining budgets $\mathbf{b}_t = (b_t^{\text{edge}}, b_t^{\text{lat}}, b_t^{\text{tok}})$. We write $s_t = (q, G_t, \mathcal{F}_t, \mathcal{P}_t, \mathbf{b}_t)$, and each agent acts on a compact observation $o_t^i = \phi_i(s_t)$.

**Action space.** We use three action families, $a_t \in \{\text{EDIT}, \text{TRAVERSE}, \text{CURATE}\}$. Let the current path prefix be $p_t = (u_0, r_1, u_1, \ldots, u_t)$ with tip $u_t$. Define the typed outgoing options $\mathcal{A}_t := \{(r, v') \in R \times V : (u_t, r, v') \in E\}$. Then,

$$
\begin{aligned}
\text{EDIT:} \quad & \{\text{ADD}(e), \text{DELETE}(e), \text{STOP}\}, & & e = (u, r, v) \in \mathcal{E}_t^{\text{cand}}, \\
\text{TRAVERSE:} \quad & \{\text{CONTINUE}(r, v'), \text{BACKTRACK}, \text{STOP}\}, & & (r, v') \in \mathcal{A}_t, \\
\text{CURATE:} \quad & \{\text{SELECT}(d), \text{STOP}\}, & & d \in \mathcal{P}_t,
\end{aligned}
$$

where $\mathcal{E}_t^{\text{cand}}$ are frontier-adjacent edges (defined below).

**Costs and budgets.** We track episode-level costs for subgraph edits, interaction steps (latency proxy), and selected tokens (prompt cost):

$$C_{\text{edge}} = \sum_t c_t^{\text{edge}}, \quad C_{\text{lat}} = \sum_t c_t^{\text{lat}}, \quad C_{\text{tok}} = \sum_t c_t^{\text{tok}},$$

with increments (at most one edit per round)

$$c_t^{\text{edge}} = \mathbf{1}\{a_t \in \{\text{ADD}, \text{DELETE}\}\}, \quad c_t^{\text{lat}} = \mathbf{1}\{a_t \neq \text{STOP}\}, \quad c_t^{\text{tok}} = \sum_{d \in \Delta D_t} \text{tok}(d),$$

where $\Delta D_t$ are newly selected units at round $t$ and $\mathrm{tok}(\cdot)$ counts tokens. Per-episode budgets are $\beta = (\beta_{\text{edge}}, \beta_{\text{lat}}, \beta_{\text{tok}})$. An episode ends when all three agents emit STOP or any budget is exhausted.

**Objective (CMDP).** Let $R_{\text{acc}}(\tau)$ be the episode reward. We solve

$$\max_{\pi} \ \mathbb{E}_{\tau \sim \pi}\big[R_{\text{acc}}(\tau)\big] \ \text{ s.t. } \ \mathbb{E}[C_{\text{edge}}] \leq \beta_{\text{edge}}, \ \ \mathbb{E}[C_{\text{lat}}] \leq \beta_{\text{lat}}, \ \ \mathbb{E}[C_{\text{tok}}] \leq \beta_{\text{tok}}. \tag{1}$$

The Lagrangian is $\mathcal{L}(\pi, \boldsymbol{\lambda}) = \mathbb{E}\big[R_{\text{acc}} - \boldsymbol{\lambda}^\top C\big]$ with prices $\boldsymbol{\lambda} = (\lambda_{\text{edge}}, \lambda_{\text{lat}}, \lambda_{\text{tok}}) \geq 0$.

**Frontierized subgraph.** A subgraph $G_t$ is *frontierized* if $\mathcal{E}_t^{\text{cand}} \subseteq \{(u, r, v) \in E : u \in \mathcal{F}_t\}$, i.e., all candidate expansions originate from the frontier. After each accepted edit or traversal, $\mathcal{F}_t$ is updated by adding touched endpoints and removing saturated nodes.

## 4 METHOD

### 4.1 CLAUSE OVERVIEW

We propose **CLAUSE**, an *agentic neuro-symbolic* framework for multi-hop KGQA that learns to *edit*, *traverse*, and *curate* compact, query-specific graph contexts under explicit per-episode budgets. An overview of the CLAUSE architecture is shown in Figure 1. CLAUSE operates over KG symbols (entities/relations/paths) with lightweight neural controllers, yielding auditable traces. Three agents act on the evolving subgraph $G_t$ and are trained *jointly* with LC–MAPPO (centralized training with a constrained multi-head critic; §4.4): (i) **Subgraph Architect** for conservative, reversible edits to keep $G_t$ compact; (ii) **Path Navigator** that decides CONTINUE/BACKTRACK/STOP along symbolic paths; and (iii) **Context Curator** that performs budget-aware evidence selection with an explicit STOP. CLAUSE exposes deployable controls via per-query budgets $(\beta_{\text{edge}}, \beta_{\text{lat}}, \beta_{\text{tok}})$ or equivalent prices $\boldsymbol{\lambda}$, enabling accuracy–efficiency trade-offs without retraining. The algorithms are given in the Appendix C.

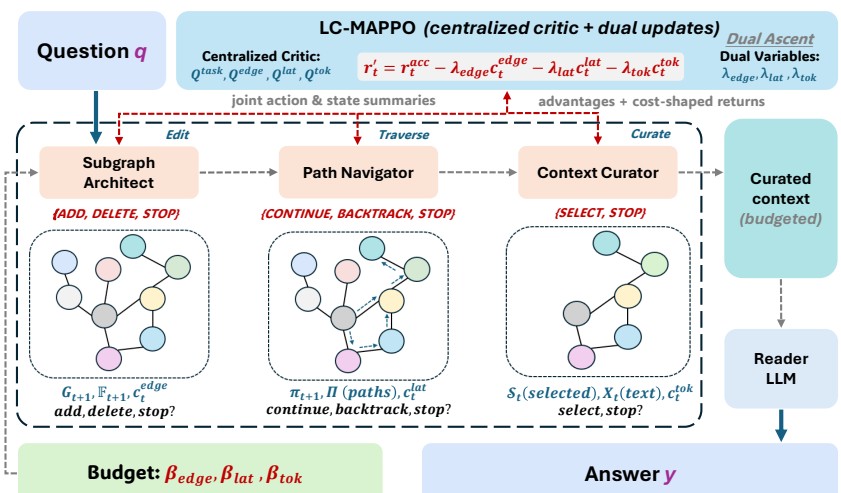

Figure 1: The CLAUSE workflow. Three agents (*Architect*, *Navigator*, *Curator*) operate on a symbolic KG state under per-episode budgets; LC–MAPPO trains task and cost heads jointly and provides deployable dials at inference.

### 4.2 DESIGN PRINCIPLES

**Per-episode budgets.** Each query carries budgets $(\beta_{\text{edge}}, \beta_{\text{lat}}, \beta_{\text{tok}})$ (or prices $\boldsymbol{\lambda}$; see §3).
**Joint control.** Editing, traversal, and curation are optimized together, replacing $k$-hop/degree/top-$k$ heuristics.
**Learned stopping.** The agents keep going only if another hop or another snippet is worth its cost; otherwise they stop.

**Neuro-symbolic transparency.** Actions are discrete KG edits/moves; neural modules provide the scores; traces are auditable.

## 4.3 AGENTIC WORKFLOW

At each decision round, CLAUSE executes a three-stage loop—*edit* $\rightarrow$ *traverse* $\rightarrow$ *curate*—conditioned on $(\beta_{\mathrm{edge}}, \beta_{\mathrm{lat}}, \beta_{\mathrm{tok}})$ or $\boldsymbol{\lambda}$. After every action, counters $(C_{\mathrm{edge}}, C_{\mathrm{lat}}, C_{\mathrm{tok}})$ update, remaining budgets are recomputed, and any agent may issue STOP; the episode ends when all modules stop or a budget is exhausted. We train three agents jointly with LC–MAPPO (Section 4.4).

**(1) Subgraph Architect (anchoring & edit).** From the question $q$, we extract mention candidates $M(q)$ by matching to entity names/aliases (optionally aided by a simple tagger). For a mention $m$ and entity $v$, we compute the anchor score $s_{\mathrm{anch}}(m, v)$. We aggregate to an entity score $s_{\mathrm{ent}}(v \mid q) = \max_{m \in M(q)} s_{\mathrm{anch}}(m, v)$, and form a seed set $S_0$ by taking the top-$k$ entities. If alias hits are weak or absent, we fall back to a frozen-encoder retrieval over entity textual fields and take the top-$k$ as seeds. The initial frontier is $\mathcal{F}_0 := S_0$; the initial subgraph $G_0$ is built around $\mathcal{F}_0$ (within budget).

Let $(G_t, \mathcal{F}_t)$ be the current subgraph and frontier $\mathcal{F}_t \subseteq V(G_t)$. The architect considers frontier-adjacent candidates $\mathcal{E}_t^{\mathrm{cand}} \subseteq \{(u, r, v) \in E : u \in \mathcal{F}_t\}$. Each candidate $e = (u, r, v)$ receives a fused score

$$s(e \mid q, G_t) = w^\top \Big[ \phi_{\mathrm{ent}}(u, q), \ \phi_{\mathrm{rel}}(r, q), \ \phi_{\mathrm{nbr}}(u, G_t), \ \phi_{\mathrm{deg}}(u) \Big],$$

where $\phi_{\mathrm{ent}}$ and $\phi_{\mathrm{rel}}$ combine lexical features and cosine similarities from a frozen encoder, and $\phi_{\mathrm{nbr}}, \phi_{\mathrm{deg}}$ encode neighborhood and degree priors (hub throttling). At step $t$ the agent chooses $a_t \in \{\text{ADD}, \text{DELETE}, \text{STOP}\}$ and, if applicable, $e_t \in \mathcal{E}_t^{\mathrm{cand}}$ to maximize the price-shaped gain

$$g(a, e \mid q, G_t) = s(e \mid q, G_t) - \lambda_{\mathrm{edge}} \, c_{\mathrm{edge}}(a, e),$$

subject to remaining edge/latency budgets. An edit is applied only if $g(a_t, e_t) > 0$ and budget remains; $(G_{t+1}, \mathcal{F}_{t+1})$ are updated accordingly. All candidates originate from the frontier, avoiding uninformed $k$-hop expansions. Per round, scoring costs $O(C_B d)$ with $C_B = |\mathcal{E}_t^{\mathrm{cand}}|$ and encoder width $d$; applying accepted edits costs $O(\Delta E)$.

**(2) Path Navigator (traverse).** Given $G_t$, the navigator maintains a path prefix $p_t$ and observes $(q, v_t, \mathcal{A}_t, \mathrm{summary}(p_t))$, where $\mathcal{A}_t$ are typed outgoing candidates. A light encoder outputs (i) a termination head over $\{\text{STOP}, \text{CONTINUE}\}$ and (ii) candidate logits over $\mathcal{A}_t$ when continuing; BACKTRACK is modeled as an explicit action. Each hop increments $C_{\mathrm{lat}}$, so continuation occurs only when expected shaped value exceeds the current step price. We cap the horizon by a small $H$ and retain log-probabilities for credit assignment. Discovered paths $\Pi = \{p_1, \ldots, p_m\}$ serve as human-readable provenance.

**(3) Context Curator (curate).** From a pool $\mathcal{P}_t$ (textualized nodes/edges/paths and optional retrieval hits), the curator performs listwise selection with an explicit STOP:

$$\max_{\pi_S} R^{\mathrm{task}}(\mathcal{S}) \quad \text{s.t.} \quad \sum_{c \in \mathcal{S}} \mathrm{tok}(c) \leq \beta_{\mathrm{tok}}, \qquad \mathcal{S} = \mathrm{Curate}(\mathcal{P}_t; \pi_S).$$

Beyond independent passage thresholds, we use *listwise, redundancy-aware* scoring with a learned STOP head *conditioned on the token price* (dual $\lambda_{\mathrm{tok}}$), aligning selection with $C_{\mathrm{tok}}$ and producing compact, complementary evidence sets that are both efficient and auditable.

**Observations and cost attribution.** Agents receive compact summaries of $(G_t, \mathcal{F}_t, \mathcal{P}_t)$ and the remaining budgets. Costs are attributed at source, edits $\rightarrow C_{\mathrm{edge}}$, steps $\rightarrow C_{\mathrm{lat}}$, curations $\rightarrow C_{\mathrm{tok}}$, which simplifies credit assignment and supplies the cost signals used by LC-MAPPO.

## 4.4 LEARNING: LC-MAPPO

To enforce per-episode budgets in *edges*, *steps*, and *tokens* while preserving accuracy, we propose LC–MAPPO, a Lagrangian-constrained CTDE variant of MAPPO that *jointly* learns task value and multiple cost processes with deployable test-time dials. A centralized critic estimates one task head $Q^{\mathrm{task}}$ and three cost heads $(Q^{\mathrm{edge}}, Q^{\mathrm{lat}}, Q^{\mathrm{tok}})$ over joint actions; a monotonic mixer aggregates per-agent utilities for each head (Rashid et al., 2020). Let $c_t^{(k)}$ denote instantaneous cost increments

whose episode sums yield $C_k$ in Eq. 1, for $k \in \{\text{edge}, \text{lat}, \text{tok}\}$. The PPO surrogate uses COMA-style counterfactual advantages (Foerster et al., 2018; Schulman et al., 2017; Yu et al., 2022) on the *shaped* return

$$r'_t = r_t^{\text{acc}} - \lambda_{\text{edge}} c_t^{\text{edge}} - \lambda_{\text{lat}} c_t^{\text{lat}} - \lambda_{\text{tok}} c_t^{\text{tok}}, \qquad (2)$$

which instantiates the *per-step Lagrangian* of the CMDP in Eq. 1. At optimum, the duals $\boldsymbol{\lambda}^\star$ equal the partial derivatives of the optimal value w.r.t. budgets (shadow-price property), and therefore predict the local slope of the accuracy–latency-cost frontiers (Appendix G).

Rather than fixing a single penalty, LC–MAPPO maintains *separate* dual variables $\lambda_{\text{edge}}, \lambda_{\text{lat}}, \lambda_{\text{tok}}$ and updates them by projected ascent,

$$\lambda_k \leftarrow \left[ \lambda_k + \eta \big( \widehat{\mathbb{E}}[C_k] - \beta_k \big) \right]_+, \quad k \in \{\text{edge}, \text{lat}, \text{tok}\},$$

optionally stabilized with PID control (Achiam et al., 2017; Stooke et al., 2020). This is stochastic dual ascent on the Lagrangian of Eq. 1, moving $\lambda$ to enforce $\mathbb{E}[C_k] \leq \beta_k$ while actors ascend the shaped objective. The separation of a task head from cost heads improves credit assignment and exposes explicit accuracy–efficiency trade-offs at test time (tune $\lambda$ or $\beta$ without retraining). Convergence is stated in Appendix H.

## 4.5 INFERENCE AND DEPLOYMENT CONTROLS

At test time, agents act greedily with learned STOP. Operators may run in *cap* mode (set $(\beta_{\text{edge}}, \beta_{\text{lat}}, \beta_{\text{tok}})$) for hard guarantees or in *price* mode (fix $\boldsymbol{\lambda}$) for smooth trade-offs—both from a single checkpoint. Symbolic decisions yield step-level traces (what was added, explored, selected, and where we stopped) for audit and ablation.

## 5 EXPERIMENTS

**Dataset.** We evaluate on three multi-hop KGQA datasets, including METAQA (Zhang et al., 2018), HOTPOTQA (Yang et al., 2018), and FACTKG (Kim et al., 2023).

**Baselines.** We compare three families under a shared retriever/reader and decoding (except the no-retrieval group). **Pretrained LLMs (no retrieval):** GPT-OSS-120B; LLaMA3.3-70B; Qwen3-32B. **RAG methods (Qwen3-32B):** Vanilla RAG (Lewis et al., 2020); Hybrid RAG (Robertson & Zaragoza, 2009; Karpukhin et al., 2020; Nogueira & Cho, 2020); LightRAG (Guo et al., 2025); GraphRAG (Edge et al., 2025). **Agent-based methods (Qwen3-32B):** ReAct (Yao et al., 2023); Graph-of-Thoughts (GoT) (Besta et al., 2024); AutoGen (Wu et al., 2023); KG-Agent (Jiang et al., 2025). Additionally, LC-MAPPO is compared with MAPPO (Yu et al., 2022), fixed-penalty PPO (Schulman et al., 2017) and single-multiplier RCPO (Tessler et al., 2019).

**Metrics. Accuracy** is reported as top-1 exact match (EM@1). **Efficiency** is measured by (i) **Average latency**, normalized so that Vanilla RAG $= 1.0\times$ per dataset/hop; and (ii) **Average edge budget**, i.e., the mean number of graph edges explored, also normalized to Vanilla RAG $= 1.0\times$.

### 5.1 EXPERIMENTAL RESULTS AND ANALYSIS

**Exact Match.** Table 2 reports EM@1 in HotpotQA (distractor), FactKG, and MetaQA. **CLAUSE** achieves the best accuracy on all datasets and hops (71.7 on HotpotQA, 84.2 on FactKG, and 91.0/87.3/85.5 on MetaQA 1/2/3-hop), consistently surpassing both RAG baselines (e.g., Hybrid RAG 66.0 on HotpotQA) and agent baselines (e.g., KG-Agent 68.7 on HotpotQA, 87.3/78.0/75.4 on MetaQA). Pure pretrained LLMs perform markedly worse, highlighting the value of budget-aware, neuro-symbolic control over subgraph editing, traversal, and evidence curation.

**Latency.** Table 3 shows average latency normalized to Vanilla RAG $= 1.0\times$. Among RAG methods, LightRAG is the fastest but sacrifices accuracy; GraphRAG is slowest because of graph construction overheads. Agent baselines incur higher latency than RAG (e.g., AutoGen and GoT are the slowest) because of multi-step tool/use deliberations. **CLAUSE** achieves *agent-level accuracy with competitive efficiency*: its latency is close to or below Hybrid/GraphRAG and substantially lower than typical agent systems (e.g., $1.48\times$ on HotpotQA vs. $2.43\times$ for AutoGen), and even dips below Vanilla on MetaQA 1-hop ($0.98\times$), reflecting effective *learned stopping* and budgeted context construction;

Table 2: Main QA results: EM@1 on HotpotQA, FactKG, and MetaQA.

| Family | Method | HotpotQA (Distractor) | FactKG | MetaQA | | |
|---|---|---|---|---|---|---|
| | | | | 1-hop | 2-hop | 3-hop |
| Pretrained-LLMs | GPT-OSS-120B | 44.5 | 68.0 | 62.7 | 41.5 | 52.3 |
| | LLaMA3.3-70B | 41.0 | 66.7 | 57.2 | 29.0 | 44.2 |
| | Qwen3-32B | 37.9 | 60.1 | 52.5 | 22.8 | 39.0 |
| RAG-based (Qwen3-32B) | Vanilla RAG | 62.1 | 77.0 | 60.2 | 37.6 | 33.0 |
| | Hybrid RAG | 66.0 | 80.2 | 63.0 | 41.5 | 34.1 |
| | LightRAG | 44.3 | 64.5 | 54.0 | 35.0 | 32.0 |
| | GraphRAG | 50.1 | 72.0 | 63.5 | 48.0 | 44.4 |
| Agent-based (Qwen3-32B) | ReAct | 63.5 | 78.2 | 82.3 | 52.1 | 49.4 |
| | Graph-of-Thoughts | 59.2 | 74.0 | 79.5 | 48.4 | 46.3 |
| | AutoGen | 64.0 | 76.5 | 85.2 | 55.7 | 53.5 |
| | KG-Agent | 68.7 | 82.1 | 87.3 | 78.0 | 75.4 |
| **Ours** | **CLAUSE** | **71.7** | **84.2** | **91.0** | **87.3** | **85.5** |

the slight rise at 2/3-hop mirrors increased multi-hop exploration while remaining well under other agentic baselines.

Table 3: Efficiency results: Average latency (normalized to Vanilla RAG = 1.0×).

| Family | Method | HotpotQA (Distractor) | FactKG | MetaQA | | |
|---|---|---|---|---|---|---|
| | | | | 1-hop | 2-hop | 3-hop |
| RAG-based (Qwen3-32B) | Vanilla RAG | 1.00 | 1.00 | 1.00 | 1.00 | 1.00 |
| | Hybrid RAG | 1.18 | 1.15 | 1.12 | 1.20 | 1.28 |
| | LightRAG | 0.85 | 0.88 | 0.80 | 0.83 | 0.86 |
| | GraphRAG | 1.45 | 1.35 | 1.25 | 1.40 | 1.60 |
| Agent-based (Qwen3-32B) | ReAct | 1.62 | 1.40 | 1.25 | 1.45 | 1.70 |
| | Graph-of-Thoughts | 2.10 | 1.78 | 1.65 | 1.90 | 2.32 |
| | AutoGen | 2.43 | 2.20 | 1.81 | 2.14 | 2.62 |
| | KG-Agent | 1.70 | 1.54 | 1.30 | 1.62 | 1.90 |
| **Ours** | **CLAUSE** | **1.48** | **1.36** | **0.98** | **1.14** | **1.27** |

**Average Edge Budget.** Table 4 reports the average edge budget normalized to Vanilla RAG (1.0×), which reflects how much the working subgraph grows during context construction. Within RAG baselines, LightRAG is the most frugal (0.75–0.82) and GraphRAG the most expansive (1.18–1.55), while Hybrid RAG sits slightly above Vanilla due to dual-channel retrieval and re-ranking. Agent systems generally consume more edges than RAG (e.g., AutoGen up to 2.10× on MetaQA-3hop) because multi-step deliberation triggers additional expansions. In contrast, **CLAUSE** achieves the smallest edge budgets across all settings (0.74–0.90) while still delivering the best EM (cf. Table 2), indicating that its budget-aware subgraph editing and learned STOP decisions effectively suppress redundant growth. The modest increase from MetaQA 1-hop to 3-hop matches the expected need to explore deeper paths, yet remains well below other agentic approaches.

Table 4: Efficiency results: Average Edge Budget (normalized to Vanilla RAG = 1.0×).

| Family | Method | HotpotQA (Distractor) | FactKG | MetaQA | | |
|---|---|---|---|---|---|---|
| | | | | 1-hop | 2-hop | 3-hop |
| RAG-based (Qwen3-32B) | Vanilla RAG | 1.00 | 1.00 | 1.00 | 1.00 | 1.00 |
| | Hybrid RAG | 1.12 | 1.08 | 1.05 | 1.12 | 1.20 |
| | LightRAG | 0.78 | 0.80 | 0.75 | 0.78 | 0.82 |
| | GraphRAG | 1.35 | 1.30 | 1.18 | 1.32 | 1.55 |
| Agent-based (Qwen3-32B) | ReAct | 1.20 | 1.13 | 1.05 | 1.18 | 1.35 |
| | Graph-of-Thoughts | 1.55 | 1.40 | 1.30 | 1.55 | 1.85 |
| | AutoGen | 1.84 | 1.72 | 1.45 | 1.75 | 2.10 |
| | KG-Agent | 1.30 | 1.22 | 1.10 | 1.32 | 1.58 |
| **Ours** | **CLAUSE** | **0.78** | **0.74** | **0.77** | **0.78** | **0.90** |

**Token Usage.** As shown in Figure 2, across all three datasets, **Qwen3-32B (no RAG)** exhibits the lowest normalized token usage (because no retrieved context is concatenated), while **CLAUSE**,

without relying on multi-agent expansion, consistently uses fewer tokens than the family averages of RAG-based and Agent-based methods, indicating better token efficiency. (Note: the MetaQA panel reports the average over the 1/2/3-hop settings.)

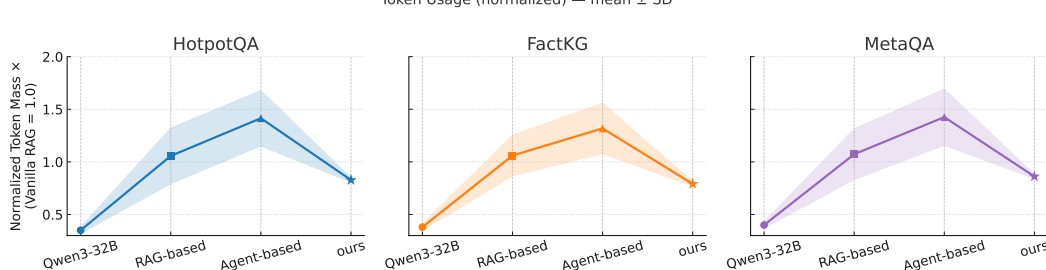

Figure 2: Normalized Token Consumption (Vanilla RAG = 1.0×).

**Constraint Satisfaction Performance.** We evaluate LC-MAPPO against MAPPO (Yu et al., 2022), Fixed-Penalty PPO (Schulman et al., 2017) and RCPO (Tessler et al., 2019) on the MetaQA KGQA task under constrained settings (edge budget = 0.5, latency budget = 0.7). Figure 3 demonstrates LC-MAPPO's superior constraint satisfaction capabilities across multiple metrics. LC-MAPPO achieves a 191% improvement in feasibility rate compared to standard MAPPO (0.340 vs. 0.117), indicating significantly better constraint adherence. Furthermore, LC-MAPPO reduces latency violations by 34% (0.577 vs. 0.880) and latency costs by 12% (0.738 vs. 0.838), demonstrating effective latency-aware optimization. LC-MAPPO demonstrates the strongest constraint learning with adaptive dual variables of 0.004, outperforming RCPO's 0.001 and surpassing methods without constraint adaptation, confirming that our multi-head centralized critic successfully learns to balance task performance with constraint satisfaction. These results validate LC-MAPPO's design for constraint-aware MARL, where the algorithm substantially improved constraint compliance.

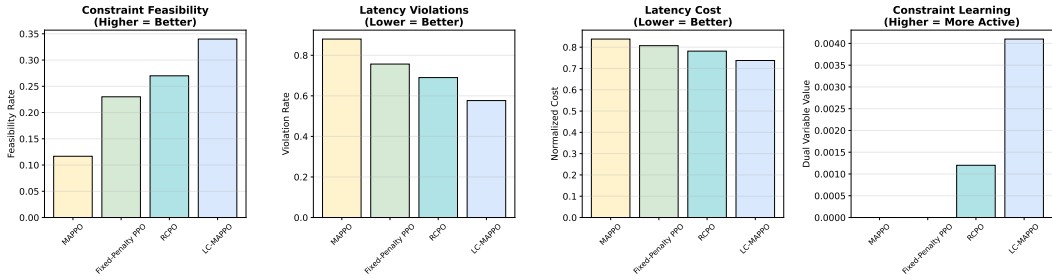

Figure 3: Constraint satisfaction performance comparison. (a) Constraint Feasibility. (b) Latency Violations. (c) Latency Cost. (d) Constraint Learning.

## 5.2 ABLATIONS

As summarized in Table 5, removing any agent or disabling constraint handling hurts both accuracy and efficiency. The full CLAUSE attains the best EM@1 (87.3) at the reference latency and edge budget (both $1.00\times$). Without the *Subgraph Architect* (StaticRAG; no-KG), EM drops sharply to 74.8 while latency and edge usage rise to $1.32\times$ and $1.44\times$, indicating severe over-expansion without budget-aware graph editing. Removing the *Path Navigator* (Greedy-Hop) yields EM 82.1 with higher latency/edges ($1.18\times/1.22\times$), showing that learned continue/backtrack/stop decisions are important for disciplined exploration. Omitting the *Context Curator* (Top-$k$ Rerank) reduces EM to 80.6 and raises latency to $1.24\times$ (edges $1.07\times$), reflecting longer, unpruned contexts when the learned stop is absent. Constraint ablations further confirm the role of LC-MAPPO: MAPPO without duals achieves EM 85.0 but overshoots edges ($1.28\times$), and fixing $\lambda$ (no updates) reaches EM 84.6 with milder but persistent budget violations ($1.06\times$ latency, $1.15\times$ edges). Together, these results demonstrate that all three agents and adaptive dual updates are necessary to jointly optimize EM, latency, and edge growth under requirements.

Table 5: Core ablations on **MetaQA**. All runs use the same reader and settings (normalized to CLAUSE = 1.0×).

| Variant | EM@1↑ | Latency↓ (avg) | Edge budget↓ (avg) |
|---|---|---|---|
| **CLAUSE (full)** | **87.3** | **1.00** | **1.00** |
| w/o Subgraph Architect  *(StaticRAG; no-KG)* | 74.8 | 1.32 | 1.44 |
| w/o Path Navigator  *(Greedy-Hop; no traversal policy)* | 82.1 | 1.18 | 1.22 |
| w/o Context Curator  *(Top-k Rerank; no learned stop)* | 80.6 | 1.24 | 1.07 |
| MAPPO *(no duals)* | 85.0 | 1.08 | 1.28 |
| Fixed $\lambda$ *(no updates)* | 84.6 | 1.06 | 1.15 |

## 5.3 CASE STUDY

> **Question:** *Who co-starred with Brian Backer?*
>
> **(1) Subgraph Architect.** Anchors: *Brian Backer* (actor).
> - **Add** `(Moving Violations, starred_actors, Brian Backer)`
> - **Add** `(...)`
> - **Stop** (edge budget nearly met)
>
> **(2) Path Navigator** Path discovered:
>
> $$\texttt{Actor\_A} \xleftarrow{\texttt{starred\_actors}} \texttt{Movie} \xrightarrow{\texttt{starred\_actors}} \texttt{Actor\_B.}$$
>
> At hop 2, backtracking is not triggered; STOP fires with high confidence due to saturated utility.
>
> **(3) Context Curator.** With $\beta_{\text{tok}} = 512$, the curator selects two snippets: "Moving Violations — starred_actors: Jennifer Tilly" and "Moving Violations — starred_actors: John Murray". Token mass is $\approx 36$ ($< \beta_{\text{tok}}$), so STOP triggers; latency is $238.6\,\text{ms}$. The reader returns *Jennifer Tilly & John Murray*. (The complete process is illustrated in Figure 4)

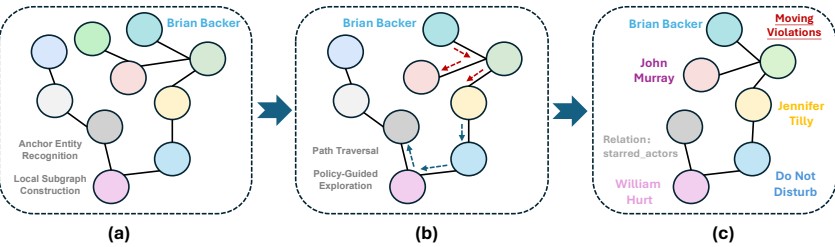

Figure 4: End-to-End Case Study Overview.

## 6 CONCLUSION

This work formulates KGQA as dynamic learnable context construction: instead of a fixed $k$-hop neighborhood, the system must decide what context to assemble, how to obtain it, and when to stop under per-query limits on edits, interaction steps, and tokens. CLAUSE instantiates this by decomposing context into three components: *(i) subgraph* structure, *(ii) path* traces, and *(iii) textual* evidence, and assigning them to three simple agents (Architect, Navigator, Curator) that make discrete, auditable choices. LC-MAPPO optimizes the overall workflow by pricing resources via a centralized multi-head critic with dual variables, so agents continue only when the predicted marginal utility exceeds the current price. This requirement-conditioned controller yields compact provenance and predictable latency/cost, and empirically traces stronger accuracy–efficiency frontiers than heuristic expansion or unconstrained agent loops.

## ETHICS STATEMENT

We comply with the ICLR Code of Ethics. This work uses only publicly available benchmark datasets and does not involve human-subject data or personally identifiable information; therefore, IRB approval was not required. The methods are intended for benign research uses and are not designed to facilitate privacy violations or discriminatory outcomes. The authors declare no conflict of interest.

## ACKNOWLEDGEMENTS

This research is supported by Seatrium New Energy Laboratory, Singapore Ministry of Education (MOE) Tier 1 (RT5/23 and RG24/24), the Nanyang Technological University (NTU) Centre for Computational Technologies in Finance (NTU-CCTF), and the Research Innovation and Enterprise (RIE) 2025 Industry Alignment Fund - Industry Collaboration Projects (IAF-ICP) (Award I2301E0026), administered by Agency for Science, Technology and Research (A*STAR).

## REPRODUCIBILITY STATEMENT

We provide an anonymized code archive in the supplementary materials. The model and objective are specified in Section 4, and pseudocode in Appendix C. Datasets and preprocessing are described in Section 5 and Appendix D. Metrics, baseline settings, and normalization are defined in Section 5; seeds, environment details, and run scripts are included in the archive.

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

APPENDIX CONTENTS

# A LARGE LANGUAGE MODELS USAGE DISCLOSURE

**Scope of use.** We used large language models in three defined roles:

1. **Writing assistance (polish/clarity only).** Micro-edits for grammar, concision, tense/voice consistency, LaTeX phrasing, section headings, figure/table captions, and title/abstract variants.

2. **Retrieval & discovery (related work support).** Query about recommended papers related to my idea.

3. **Research ideation (early brainstorming).** Generating alternative task framings, evaluating feasibility of this, naming options for modules.

**Author responsibility.** The authors are solely responsible for the methods, experiments, analyses, and claims. LLMs supported drafting, search query design, and ideation; they did not generate or select results.

# B  NOTATION TABLE

Table 6: Symbols and notation used throughout the paper.

| Symbol | Description |
|---|---|
| $\mathcal{K} = (V, R, E)$ | Global knowledge graph with entities $V$, relations $R$, and edges $E \subseteq V \times R \times V$ |
| $G_t = (V_t, E_t); G^*$ | Query-conditioned subgraph at step $t$; final subgraph at termination |
| $\mathcal{F}_t$ | Frontier node set at step $t$ |
| $\mathcal{P}_t$ | Candidate pool (textualized nodes/edges/paths and optional retrieval hits) at step $t$ |
| $\mathcal{A}_t$ | Typed outgoing candidates (actions) available to the navigator at step $t$ |
| $q; y, \hat{y}$ | Input question; gold / predicted answers |
| $p_t; \Pi = \{p_1, \ldots, p_m\}$ | Path prefix maintained by the navigator; set of discovered paths (provenance) |
| $\pi_B, \pi_T, \pi_S$ | Policies for EDIT (Subgraph Architect), TRAVERSE (Path Navigator), CURATE (Context Curator) |
| $a_t^B, a_t^T, a_t^S$ | Actions at step $t$ for edit / traverse / curate modules |
| $s_t$ | State summary at step $t$: $(q, G_t, \mathcal{F}_t, \mathcal{P}_t, \mathbf{b}_t)$ |
| $\mathbf{b}_t$ | Remaining budget vector at step $t$ |
| $\beta = (\beta_{\text{edge}}, \beta_{\text{lat}}, \beta_{\text{tok}})$ | Episode budget vector for edge growth, latency (steps), and selected tokens |
| $C = (C_{\text{edge}}, C_{\text{lat}}, C_{\text{tok}})$ | Cumulative costs (accepted edits / interaction steps / selected tokens) |
| $c_t^{(k)}$ | Instantaneous cost increment at step $t$ for $k \in \{\text{edge}, \text{lat}, \text{tok}\}$; $\sum_t c_t^{(k)} = C_k$ |
| $\boldsymbol{\lambda} = (\lambda_{\text{edge}}, \lambda_{\text{lat}}, \lambda_{\text{tok}})$ | Lagrange multipliers (prices) for edge/step/token costs |
| $R_{\text{acc}}(\tau), r_t^{\text{acc}}, r_t'$ | Episode-level task reward, per-step (unshaped) task reward, and shaped return $r_t' = r_t^{\text{acc}} - \sum_k \lambda_k c_t^{(k)}$ |
| $\mathcal{L}(\pi, \boldsymbol{\lambda})$ | Lagrangian objective $\mathbb{E}[R_{\text{acc}}(\tau) - \boldsymbol{\lambda}^\top C(\tau)]$ for the CMDP |
| $Q^{\text{task}}, Q^{\text{edge}}, Q^{\text{lat}}, Q^{\text{tok}}$ | Centralized-critic action-values (task head and three cost heads) |
| $A_t^{i,h}, A_t^{i,\lambda}; i \in \{B, T, S\}$ | Counterfactual advantage for agent $i$ and head $h$; Lagrangian-shaped advantage for PPO |
| $D^\star$ | Ordered, curated evidence list passed to the reader |
| $\text{tok}(\cdot)$ | Token count operator for a textual unit |
| $e = (u, r, v)$ | KG edge (triple) with head entity $u$, relation $r$, tail entity $v$ |
| $\mathcal{E}_t^{\text{cand}}$ | Frontier-adjacent candidate edges considered by the Architect at step $t$ |
| $s(e \mid q, G_t)$ | Fused edge-utility score used by the Architect for edit decisions |
| $H, \bar{d}, \lvert\mathcal{P}_t\rvert, K$ | Hop cap; average local branching factor; candidate pool size; selected list length ($K$ dynamic) |

## C ALGORITHM

---

**Algorithm 1** CLAUSE: Training with a constrained centralized-critic PPO (LC-MAPPO)

---

1: **Input:** dataset $\mathcal{D}$ of (question, answer) pairs; budgets $\beta = \{\beta_{\text{edge}}, \beta_{\text{lat}}, \beta_{\text{tok}}\}$; learning rates.
2: **Initialize:** actor params for EDIT/TRAVERSE/CURATE $(\theta_B, \theta_T, \theta_S)$; centralized critic $\psi$ with multi-head outputs $\{Q^{\text{task}}, Q^{\text{edge}}, Q^{\text{lat}}, Q^{\text{tok}}\}$; duals $\lambda = \{\lambda_{\text{edge}}, \lambda_{\text{lat}}, \lambda_{\text{tok}}\} \geq 0$. epoch $= 1, \ldots, E$
3: **Rollout.** Sample a minibatch $\mathcal{B} \subset \mathcal{D}$. For each question $q \in \mathcal{B}$:
4:      Reset $G_0, \mathcal{F}_0, \mathcal{P}_0$, budgets $b_0$, buffer $\mathcal{T} \leftarrow \varnothing$; $t \leftarrow 0$. NOT_TERMINAL and $t < T_{\max}$
5:      $a_t^B \sim \pi_B(\cdot \mid s_t)$; apply edit (add/delete/stop) to $G_t$; accrue $c_t^{\text{edge}}$.
6:      $a_t^T \sim \pi_T(\cdot \mid s_t)$; continue or STOP exploration; accrue $c_t^{\text{lat}}$.
7:      $a_t^S \sim \pi_S(\cdot \mid s_t)$; curate context or STOP; accrue $c_t^{\text{tok}}$.
8:      Log $(s_t, \mathbf{a}_t, \log \pi_t, c_t)$ to $\mathcal{T}$; update state $s_{t+1}$; $t \leftarrow t+1$.
9:      Produce answer $\hat{y}$; compute terminal task reward $r^{\text{acc}}$; finalize episode $\mathcal{T}$.
10: **Critic update.** Compute targets with GAE/TD for each head using $\mathcal{T}$; update $\psi$ (task & cost heads).
11: **Advantages.** Form shaped rewards $r_t' = r_t^{\text{acc}} - \lambda_{\text{edge}} c_t^{\text{edge}} - \lambda_{\text{lat}} c_t^{\text{lat}} - \lambda_{\text{tok}} c_t^{\text{tok}}$; compute GAE; for each agent $i \in \{B, T, S\}$, compute *counterfactual* advantages

$$A_i(s_t, \mathbf{a}_t) = Q^{\text{task}}(s_t, \mathbf{a}_t) - \mathbb{E}_{a_i'}\big[Q^{\text{task}}(s_t, (\mathbf{a}_{-i}, a_i'))\big].$$

12: **Actor update.** Apply PPO to $\{\theta_B, \theta_T, \theta_S\}$ with clipped ratios, entropy bonus, and using the agent-specific $A_i$.
13: **Dual update.** For each $k \in \{\text{edge}, \text{lat}, \text{tok}\}$:

$$\lambda_k \leftarrow \big[\lambda_k + \rho\big(\widehat{\mathbb{E}}[C_k] - \beta_k\big)\big]_+.$$

---

**Algorithm 2** Budgeted Inference (CLAUSE, decentralized execution)

---

1: **Input:** question $q$; global budgets $\beta = \{\beta_{\text{edge}}, \beta_{\text{lat}}, \beta_{\text{tok}}\}$; trained agents $\pi_B$ (Subgraph Architect), $\pi_T$ (Path Navigator), $\pi_S$ (Context Curator).
2: Initialize $G_0 \leftarrow \varnothing$; frontier $\mathcal{F}_0 \leftarrow \text{anchors}(q)$; pool $\mathcal{P}_0 \leftarrow \varnothing$; counters $e=0, \ell=0, \tau=0$. $\ell < \beta_{\text{lat}}$ **and** not STOPPED
3: $a^B \sim \pi_B(\cdot \mid q, G_t, \mathcal{F}_t, \beta-\text{usage})$ {Add/Delete/Stop edits}
4: Apply $a^B$; update $G_t, \mathcal{F}_t$; $e \leftarrow e + \text{edit\_count}(a^B)$ $e > \beta_{\text{edge}}$ **break**
5: $a^T \sim \pi_T(\cdot \mid q, G_t, \mathcal{F}_t, \beta-\text{usage})$ {Next hop or STOP}
6: Apply $a^T$; update traversal traces; push touched edges/nodes into candidate pool $\mathcal{P}_t$
7: $a^S \sim \pi_S(\cdot \mid q, \mathcal{P}_t, \beta-\text{usage})$ {Select or STOP}
8: Update selected set $D^\star$ and token mass $\tau$; **if** $\tau > \beta_{\text{tok}}$ **then** prune last add or force STOP
9: $\ell \leftarrow \ell + 1$
10: **return** reader answer $\hat{y} \leftarrow \text{Reader}(q, D^\star)$ and the trace $\langle G^*, \text{paths}, D^\star \rangle$

---

**Algorithm 3** Counterfactual Advantage (per head) with Centralized Critic

---

1: **Given:** critic heads $\{Q^h\}$ for $h \in \{\text{task}, \text{edge}, \text{lat}, \text{tok}\}$; joint action $\mathbf{a}_t$; per-agent agents $\{\pi_i\}$ agent $i \in \{\text{B, T, S}\}$ head $h$
2: $b^{i,h}(s_t, \mathbf{a}_t^{-i}) \leftarrow \mathbb{E}_{a_i \sim \pi_i(\cdot \mid o_t^i)}\big[Q^h(s_t, (a_i, \mathbf{a}_t^{-i}))\big]$
3: $A_t^{i,h} \leftarrow Q^h(s_t, \mathbf{a}_t) - b^{i,h}(s_t, \mathbf{a}_t^{-i})$
4: Lagrangian advantage: $A_t^{i,\text{lag}} \leftarrow A_t^{i,\text{task}} - \lambda_{\text{edge}} A_t^{i,\text{edge}} - \lambda_{\text{lat}} A_t^{i,\text{lat}} - \lambda_{\text{tok}} A_t^{i,\text{tok}}$
5: **return** $\{A_t^{i,\text{lag}}\}$

---

---

**Algorithm 4** Context Curator (list-wise, dynamic $K$) — one pass

---

1: **Input:** question $q$, candidate pool $\mathcal{P}$ with token counts $\text{tok}(\cdot)$, budget $\beta_{\text{tok}}$
2: Initialize $D^\star \leftarrow \langle\rangle$, used_tokens $T \leftarrow 0$, remaining $\mathcal{R} \leftarrow \mathcal{P} \; \mathcal{R} \neq \varnothing$
3: Score all $d \in \mathcal{R}$ with list-wise scorer $s(d \mid q, \mathcal{R}, D^\star)$
4: $d^\star \leftarrow \arg\max_{d \in \mathcal{R}} s(d) \; T + \text{tok}(d^\star) > \beta_{\text{tok}}$ **break**
5: Append $d^\star$ to $D^\star$; $T \leftarrow T + \text{tok}(d^\star)$; $\mathcal{R} \leftarrow \mathcal{R} \setminus \{d^\star\}$
6: With probability $\pi_S(\text{STOP} \mid q, \mathcal{R}, D^\star, \beta-\text{usage})$ do **break**
7: **return** $D^\star$

---

## D  IMPLEMENTATION DETAILS & SUPPLEMENTARY EXPERIMENT

**Hardware.** All experiments are run on a single NVIDIA L20 (48 GB) GPU with 20 vCPU Intel Xeon Platinum 8457C, 100 GB RAM, and two SSD partitions (system: 128 GB; data: 1024 GB). The OS image is `Ubuntu 22.04` with `Miniconda (Python 3.10)`.

**Software & API.** We use PyTorch (CUDA 11.8 build). Retrieval *does not* use a generative LLM: we produce dense embeddings with sentence-transformers (`thenlper/gte-small`) and perform lightweight NER with *spaCy*; sparse recall uses BM25. Vectors are L2-normalized and queried with FAISS `IndexFlatIP` (cosine-equivalent). Unless stated otherwise, the *shared reader LLM* for all RAG/agent baselines is `Qwen3-32B`. We also evaluate `GPT-OSS-120B` and `LLaMA3.3-70B` for ablations/comparisons and final answer generation, invoked via SiliconFlow / Groq APIs. Token budgets and token counting always use the *reader's* tokenizer.

**Datasets & Knowledge Graphs (KGs).** We test CLAUSE on three large-scale, real-world, noisy KGs, and provide construction details. We do *not* share a single KG across datasets; each benchmark uses its own canonical KG (Table 7). Our graphs are cross-domain: METAQA relies on a movie/entertainment-domain KB, HOTPOTQA uses a Wikidata-derived subgraph built from general encyclopedic entities, and FACTKG is grounded in the DBpedia open-domain fact graph. Anchoring links surface mentions to KG entities via NER and an alias dictionary; ties or unresolved mentions fall back to dense nearest neighbors in the alias embedding space. For HOTPOTQA, we first run entity recognition and alias matching over the questions and supporting paragraphs, link mentions to Wikidata entities, and then extract a local neighborhood (within a fixed hop radius) for these entities from a frozen Wikidata dump, yielding a subgraph with roughly millions of nodes and tens of millions of edges. In Table 7, we report the scale of this subgraph using $\mathcal{O}(10^{\cdot})$ notation because the exact node/edge counts can vary slightly with the chosen Wikidata version and neighborhood radius, while the order of magnitude is sufficient to convey that the setting is web-scale.

Table 7: Benchmarks and knowledge graphs. We report the KG identity, scale, and the exact anchoring protocol. All RAG/agent baselines share the same reader and tokenizer.

| Benchmark | KG source | #Nodes | #Edges | #Relations |
|---|---|---|---|---|
| METAQA (1/2/3-hop) | (Zhang et al., 2018) | $4.3 \times 10^4$ | $1.35 \times 10^5$ | 9 |
| HOTPOTQA (distractor) | (Yang et al., 2018) | $\mathcal{O}(10^6)$ | $\mathcal{O}(10^7)$ | $\mathcal{O}(10^3)$ |
| FACTKG | (Kim et al., 2023) | $\sim 4 \times 10^6$ | $\sim 1 \times 10^8$ | $\sim 10^3$ |

**Default inference settings.** Unless stated otherwise, we adopt the following default configuration: the token budget is set to $\beta_{\text{tok}} = 512$; the reranker runs for 15 steps with a minimum of $K_{\text{min}} = 2$ picks; the traversal is capped at 4 hops (with an earlier stop if the learned stopping policy fires); and the curator operates on a candidate pool of size 128 with a dense-fallback pool of 64 candidates.

**Supplementary experiments with alternative LLM backbones.** We further report results when using two alternative reader LLM backbones, LLaMA3.3-70B and GPT-OSS-120B, for all RAG/agent baselines and CLAUSE. The detailed numerical results are given in Tables 8 and 9. Across all three backbones (Qwen3-32B, LLaMA3.3-70B, and GPT-OSS-120B), CLAUSE consistently achieves the best EM on HotpotQA, FactKG, and MetaQA. At the same time, the absolute gap between different backbones becomes much smaller once they are coupled with our budget-aware neuro-symbolic controller: Qwen3-32B+CLAUSE already matches or nearly matches the performance of larger backbones, indicating that CLAUSE can largely close the performance gap between medium and large LLMs while keeping the reader size flexible.

Table 8: Main QA results: EM@1 on HotpotQA, FactKG, and MetaQA using LLaMA3.3-70B.

| Family | Method | HotpotQA (Distractor) | FactKG | MetaQA | | |
|---|---|---|---|---|---|---|
| | | | | 1-hop | 2-hop | 3-hop |
| RAG-based (LLaMA3.3-70B) | Vanilla RAG | 63.8 | 78.3 | 62.0 | 38.6 | 45.4 |
| | Hybrid RAG | 66.7 | 82.0 | 64.3 | 43.0 | 45.8 |
| | LightRAG | 48.6 | 68.2 | 59.6 | 36.8 | 45.0 |
| | GraphRAG | 51.4 | 73.4 | 65.0 | 49.5 | 50.3 |
| Agent-based (LLaMA3.3-70B) | ReAct | 65.0 | 79.5 | 83.0 | 53.3 | 50.6 |
| | Graph-of-Thoughts | 60.3 | 75.2 | 81.0 | 49.0 | 47.6 |
| | AutoGen | 65.2 | 77.3 | 86.2 | 56.5 | 55.1 |
| | KG-Agent | 69.3 | 83.3 | 88.0 | 78.8 | 76.3 |
| **Ours** | **CLAUSE** | **73.5** | **85.0** | **92.7** | **87.7** | **87.2** |

Table 9: Main QA results: EM@1 on HotpotQA, FactKG, and MetaQA using GPT-OSS-120B.

| Family | Method | HotpotQA (Distractor) | FactKG | MetaQA | | |
|---|---|---|---|---|---|---|
| | | | | 1-hop | 2-hop | 3-hop |
| RAG-based (GPT-OSS-120B) | Vanilla RAG | 65.9 | 79.2 | 64.8 | 44.9 | 52.5 |
| | Hybrid RAG | 68.9 | 82.9 | 67.0 | 49.2 | 52.8 |
| | LightRAG | 50.7 | 69.2 | 62.9 | 43.0 | 52.4 |
| | GraphRAG | 53.5 | 74.3 | 67.8 | 55.6 | 54.3 |
| Agent-based (GPT-OSS-120B) | ReAct | 65.5 | 80.6 | 84.1 | 65.5 | 53.5 |
| | Graph-of-Thoughts | 61.0 | 76.0 | 81.7 | 49.3 | 52.6 |
| | AutoGen | 66.6 | 78.3 | 87.0 | 66.0 | 55.8 |
| | KG-Agent | 69.8 | 84.2 | 88.9 | 79.4 | 76.9 |
| **Ours** | **CLAUSE** | **75.0** | **85.8** | **92.2** | **88.0** | **87.5** |

# E  COMPLEXITY ANALYSIS

## E.1  TIME COMPLEXITY

Let $|V^*|$, $|E^*|$ be the sizes of the constructed subgraph, $b$ the average out-degree on active frontiers, $H$ the hop cap, $P$ the candidate-pool size, $K$ the selected context length (dynamic), and $d$ the embedding width of light encoders.

**Subgraph Architect (anchoring & edit).** Each edit round ranks at most $C_B$ candidate edges (retrieval + local expansion). With precomputed entity/relation embeddings and cached neighbor lists, scoring is $O(C_B d)$; applying batched edits is $O(\Delta E)$ where $\Delta E$ is the number of accepted edits. Over $T_B$ rounds,

$$T_{\text{edit}} = O\left(T_B \cdot C_B d + \sum_t \Delta E_t\right) \subseteq O(\beta_{\text{lat}} \cdot C_B d + \beta_{\text{edge}}).$$

**Path Navigator (traverse).** At each hop, neighborhood scoring is $O(bd)$ (or $O(\deg(v))$ with simple degree-based features). For $T_T \leq H$ hops and small beam $B_{\text{beam}}$:

$$T_{\text{traverse}} = O(T_T \cdot B_{\text{beam}} \cdot bd) \subseteq O(\beta_{\text{lat}} \cdot bd).$$

**Context Curator (curate).** A single-pass pointer/list-wise scorer over $P$ items costs $O(Pd)$ per step; selecting $K$ items without replacement:

$$T_{\text{curate}} = O(K \cdot Pd) \quad \text{(naive)}, \qquad O(Pd + K \log P) \quad \text{(with heap/top-}K\text{)}.$$

Given a token budget $\beta_{\text{tok}}$ and average item length $\bar{t}$, $K \leq \beta_{\text{tok}}/\bar{t}$.

**Reader call.** The reader sees only the curated $K$ items with token mass $\leq \beta_{\text{tok}}$. Reader runtime is thus $O(\beta_{\text{tok}})$ (with a fixed decoder).

**Training cost (LC-MAPPO).** Let $S$ be steps per trajectory, $M$ mini-batch size of trajectories, and #heads $= 4$ (task/edge/lat/tok). The critic's forward/backward per update is $O(MSd \cdot \#\text{heads})$; actor updates add $O(MS|\mathcal{A}_i|)$ across the three agents. Because costs are attributed at source, the critic's label construction is linear in $S$ with negligible overhead.

## E.2 SPACE COMPLEXITY

The dominant terms are (i) subgraph cache $O(|V^*| + |E^*|)$, (ii) candidate pool $O(P)$, and (iii) replay/rollout buffers $O(MS)$ during training. Budgets bound $|E^*|$ and $S$ by $\beta_{\text{edge}}$ and $\beta_{\text{lat}}$ respectively.

**End-to-end bound under budgets.** With explicit budgets,

$$T_{\text{end2end}} = \underbrace{O(\beta_{\text{lat}} C_B d + \beta_{\text{edge}})}_{\text{edit}} + \underbrace{O(\beta_{\text{lat}} bd)}_{\text{traverse}} + \underbrace{O(Pd + K \log P)}_{\text{curate}} + \underbrace{O(\beta_{\text{tok}})}_{\text{reader}}$$

so latency scales linearly in $\beta_{\text{lat}}$ and $\beta_{\text{tok}}$, allowing direct runtime control.

## F  TRAINING COST AND SCALABILITY TO LARGE AND NOISY KGS

Let $m=3$ agents (Architect, Navigator, Curator), $H_{\text{heads}}=4$, batch $(M, S)$, and width $d$. One update decomposes as

$$T_{\text{update}} = T_{\text{env}}(M, S) + T_{\text{actor}}(M, S) + T_{\text{critic}}(M, S) + T_{\text{io/cache}}.$$

$T_{\text{env}}$ scales with edge scoring and neighbor expansions (counts $\sum_t C_B$ and $\sum_t b_t$), alias ANN fallbacks, and accepted edits $C_{\text{edge}}$. $T_{\text{actor}}$ covers $\approx MS$ forward passes per agent with candidate sizes $(C_B, b, P)$; $T_{\text{critic}}$ uses $MS$ samples with $H_{\text{heads}}$ and a GAE horizon; $T_{\text{io/cache}}$ captures FAISS lookups and adjacency fetches. The three-agent CTDE design does not triple cost: encoders are shared for representation reuse across settings (small per-agent MLP heads) (Zhou, 2025), action sets are typed and small (Architect: $C_B \ll |E|$; Navigator: $b$; Curator: pruned $P$), and a single centralized critic has $H_{\text{heads}}$ heads. Thus, constants grow with $m$ and $H_{\text{heads}}$, while dependence remains linear in $S$ and tokens. A per-episode spend model is

$$\text{Cost} \approx c_{\text{sym}} \cdot \left( \beta_{\text{lat}} C_B d + \beta_{\text{edge}} + \beta_{\text{lat}} b d + P d + K \log P \right) + c_{\text{LLM}} \cdot \beta_{\text{tok}},$$

with $\beta_{\text{tok}}$ fixed/small and the symbolic term linear in the budgets, yielding bounded and predictable cost.

For memory and indexing, entity/alias embeddings and FAISS live on CPU; GPU holds the active subgraph $G_t$ and batched activations:

$$\text{RAM} \approx O(|V| d) + O(\text{FAISS}) + O(|V^*|+|E^*|) + O(MS).$$

Product quantization or 8-bit embeddings reduce ANN footprint by $\times 3$–$\times 8$. For large KGs, frontier-ization and budgeted pricing make per-step work depend on local branching ($b$) and $(\beta_{\text{edge}}, \beta_{\text{lat}}, \beta_{\text{tok}})$ rather than $|E|$: we use IVF+HNSW or IVF+PQ for alias lookup, degree/type-aware neighbor sampling to cap $C_B$ and $b$, partitioned/frontier caches, and hub pricing via a degree prior in $s(e \mid q, G_t)$. Noise is mitigated via price-aware acceptance $s(e) - \lambda_{\text{edge}} c_{\text{edge}} > 0$, listwise de-duplication, and type/logic constraints. For reproducibility, report per epoch: $(M, S)$ and total env steps; feasible rate under $(\beta_{\text{edge}}, \beta_{\text{lat}}, \beta_{\text{tok}})$; seconds/update split (env/actor/critic/I/O); averages of $C_B$, $b$, $P$, $K$; FAISS ms/query and cache hit rate; peak CPU/GPU RAM; index size; and average reader tokens.

# G  TRADE-OFF ANALYSIS

## G.1  LAGRANGIAN SHAPING: COSTS AS PRICES, DUALS AS SHADOW VALUES

Recall the constrained objective (Eq. 1) $\max_\pi \ \mathbb{E}[R_{\mathrm{acc}}(\tau)]$ s.t. $\mathbb{E}[C_k(\tau)] \leq \beta_k, \ k \in \{\text{edge}, \text{lat}, \text{tok}\}$. The Lagrangian is $\mathcal{L}(\pi, \lambda) = \mathbb{E}\big[R_{\mathrm{acc}} - \sum_k \lambda_k(C_k - \beta_k)\big]$, and the actor updates optimize the *shaped* return $R' = R_{\mathrm{acc}} - \sum_k \lambda_k C_k$ (Eq. 2) while dual ascent updates $\lambda$ to enforce the constraints.

**Proposition 1** (Shadow-price interpretation). *Let $J^\star(\beta)$ be the optimal value of the CMDP as a function of the budget vector $\beta$. Under standard regularity, the optimal duals $\lambda^\star$ satisfy the envelope property $\frac{\partial J^\star(\beta)}{\partial \beta_k} = \lambda_k^\star$. Hence, $\lambda_k^\star$ is the* marginal EM gain per unit relaxation *of budget $k$ (edge, latency, or tokens).*

*Sketch.* This is a standard consequence of the envelope theorem for convex CMDPs and KKT conditions (Appendix H). ☐

**Practical reading.**  $\lambda_{\mathrm{lat}}$ quantifies the EM increase obtainable by allowing one more interaction step on average; $\lambda_{\mathrm{tok}}$ quantifies EM gain per additional token; $\lambda_{\mathrm{edge}}$ quantifies the benefit of expanding the subgraph. Increasing any $\lambda_k$ acts like raising the price of that resource, biasing policies toward using less of it.

# H    PROOF OF CONVERGENCE OF LC-MAPPO

This appendix states conditions under which the proposed LC-MAPPO converges to a stationary point of the CMDP Lagrangian and satisfies the KKT conditions. We follow the two/three–time–scale view in the assumptions (A1)–(A6): the centralized critic tracks action-values (fast), the actor ascends the (penalized) objective (medium), and the dual variables perform projected ascent on constraint violation (slow).

## H.1    ASSUMPTIONS

**A1 (Bounded Rewards and Costs).** There exists $C_r > 0$ such that

$$|r(s,a)| \leq C_r, \qquad |c_k(s,a)| \leq C_r, \quad \forall (s,a),\, k.$$

Thus $J_r(\theta)$ and $J_{c_k}(\theta)$ are finite and admit stochastic gradients with bounded second moments.

**A2 (Smoothness).** For any fixed dual vector $\lambda$, the penalized objective

$$F(\theta; \lambda) = J_r(\theta) - \sum_k \lambda_k \left( J_{c_k}(\theta) - \beta_k \right)$$

is $L_F$-smooth in $\theta$:

$$\|\nabla_\theta F(\theta; \lambda) - \nabla_\theta F(\theta'; \lambda)\| \leq L_F \|\theta - \theta'\|.$$

**A3 (Asymptotic Unbiasedness).** The stochastic policy gradient estimator $\hat{g}_t$ satisfies

$$\mathbb{E}[\hat{g}_t \mid \mathcal{F}_t] = \nabla_\theta F(\theta_t; \lambda) + \varepsilon_t, \qquad \varepsilon_t \to 0.$$

**A4 (Bounded Variance).** There exists $C_g > 0$ such that

$$\mathbb{E}\|\hat{g}_t\|^2 \leq C_g, \qquad \forall t.$$

**A5 (Step-Size and Time-Scale Separation).** The actor step-size $\{\alpha_t\}$ satisfies

$$\sum_{t=0}^{\infty} \alpha_t = \infty, \qquad \sum_{t=0}^{\infty} \alpha_t^2 < \infty.$$

The dual step-size $\rho_t$ is slower, i.e., $\rho_t \ll \alpha_t$. All stochastic gradients admit bounded second moments.

**A6 (Small Trust-Region Bias).** Let $\hat{\theta}_{t+1} = \theta_t - \alpha_t g_t$ be the ideal gradient step. The actual PPO/COMA update $\theta_{t+1}$ satisfies

$$|F(\theta_{t+1}; \lambda) - F(\hat{\theta}_{t+1}; \lambda)| \leq C_b \alpha_t^2.$$

## H.2    PRIMAL (ACTOR) CONVERGENCE FOR FIXED DUALS

Fix $\lambda$ and define the *penalized* objective $L(\theta, \lambda) = J_R(\theta) - \sum_k \lambda_k \left( J_{C_k}(\theta) - \beta_k \right)$, and its minimization surrogate

$$F(\theta; \lambda) \triangleq -L(\theta, \lambda). \tag{3}$$

Assume $F(\cdot; \lambda)$ is $L_\theta$-smooth in $\theta$ (from (A2) and bounded rewards/costs (A1)). Denote by $g_t = \nabla_\theta F(\theta_t; \lambda)$ the exact gradient and by $\hat{g}_t$ the PPO/GAE/COMA-based stochastic gradient used by LC-MAPPO. The actor step is

$$\theta_{t+1} = \theta_t - \alpha_t \hat{g}_t, \tag{4}$$

with $\{\alpha_t\}$ satisfying (A5). Let the *ideal* step be $\hat{\theta}_{t+1} \triangleq \theta_t - \alpha_t g_t$. By the $L_\theta$-smoothness (second-order Taylor bound),

$$F(\theta_{t+1}; \lambda) \leq F(\hat{\theta}_{t+1}; \lambda) + \left\langle \theta_{t+1} - \hat{\theta}_{t+1}, \nabla F(\hat{\theta}_{t+1}; \lambda) \right\rangle + \frac{L_\theta}{2} \|\theta_{t+1} - \hat{\theta}_{t+1}\|^2. \tag{5}$$

Taking expectations conditioned on the filtration $\mathcal{F}_t$ (all randomness up to $t$) and using the surrogate's (asymptotic) unbiasedness (A3)–(A4) and small-trust-region bias (A6),

$$\mathbb{E}[F(\theta_{t+1}; \lambda) \,|\, \mathcal{F}_t] \leq \mathbb{E}\left[F(\widehat{\theta}_{t+1}; \lambda) \,\Big|\, \mathcal{F}_t\right] + \frac{L_\theta}{2} \mathbb{E}\left[\|\theta_{t+1} - \widehat{\theta}_{t+1}\|^2 \,\Big|\, \mathcal{F}_t\right]$$
$$\overset{(a)}{=} F(\widehat{\theta}_{t+1}; \lambda) + \frac{L_\theta}{2} \alpha_t^2 \,\mathbb{E}\left[\|\widehat{g}_t - g_t\|^2 \,\big|\, \mathcal{F}_t\right], \tag{6}$$

where $(a)$ uses $\theta_{t+1} - \widehat{\theta}_{t+1} = -\alpha_t(\widehat{g}_t - g_t)$ and $\mathbb{E}[\widehat{g}_t \,|\, \mathcal{F}_t] = g_t + \varepsilon_t$ with $\|\varepsilon_t\| \leq c_{\text{TR}}\epsilon\bar{D}_{\text{KL}} + c_Q\delta_Q$ capturing trust-region and critic-tracking biases (constants from (A3),(A6)). Next, smoothness around $\theta_t$ gives

$$F(\widehat{\theta}_{t+1}; \lambda) \;\leq\; F(\theta_t; \lambda) + \left\langle \widehat{\theta}_{t+1} - \theta_t, \nabla F(\theta_t; \lambda) \right\rangle + \frac{L_\theta}{2} \|\widehat{\theta}_{t+1} - \theta_t\|^2, \tag{7}$$

which, using $\widehat{\theta}_{t+1} - \theta_t = -\alpha_t g_t$, yields

$$F(\widehat{\theta}_{t+1}; \lambda) \;\leq\; F(\theta_t; \lambda) - \alpha_t\|g_t\|^2 + \frac{L_\theta}{2}\alpha_t^2\|g_t\|^2. \tag{8}$$

Combining Eq. 6 and Eq. 8 and taking the total expectation,

$$\mathbb{E}[F(\theta_{t+1}; \lambda)] - \mathbb{E}[F(\theta_t; \lambda)] \leq -\alpha_t \,\mathbb{E}\left[\|g_t\|^2\right] + \frac{L_\theta}{2}\alpha_t^2 \,\mathbb{E}\left[\|g_t\|^2\right] + \frac{L_\theta}{2}\alpha_t^2 \,\mathbb{E}\left[\|\widehat{g}_t - g_t\|^2\right]$$
$$\leq -\frac{\alpha_t}{2} \,\mathbb{E}\left[\|g_t\|^2\right] + C_1\,\alpha_t^2 + C_2\,\alpha_t^2\left(\epsilon^2\bar{D}_{\text{KL}}^2 + \delta_Q^2\right), \tag{9}$$

for bounded second moments (A5) and small enough $\alpha_t$ so that $1 - \frac{L_\theta}{2}\alpha_t \geq \frac{1}{2}$. Summing Eq. 9 over $t$ and using $\sum_t \alpha_t = \infty$, $\sum_t \alpha_t^2 < \infty$ (A5), Robbins–Siegmund implies

$$\sum_{t=0}^{\infty} \alpha_t \,\mathbb{E}\left[\|g_t\|^2\right] < \infty \quad\Longrightarrow\quad \liminf_{t\to\infty} \mathbb{E}\left[\|\nabla_\theta F(\theta_t; \lambda)\|^2\right] = 0. \tag{10}$$

Hence any limit point $\theta^\star(\lambda)$ is stationary for $F(\cdot; \lambda)$, equivalently stationary for $L(\cdot, \lambda)$.

## H.3 DUAL (MULTIPLIER) CONVERGENCE AND KKT

The dual update is

$$\lambda_{k,t+1} \;=\; \left[\lambda_{k,t} + \rho_t\left(\widehat{J}_{C_k}(\theta_t) - \beta_k\right)\right]_+, \tag{11}$$

with $\rho_t$ satisfying (A5) and $\rho_t \ll \alpha_t$ (actor faster than duals). On the slower time scale, the actor tracks $\theta^\star(\lambda_t)$ from Part A, so the dual recursion behaves as a projected gradient ascent on the concave dual function $D(\lambda) = \max_\theta L(\theta, \lambda)$. Standard two–time–scale results then yield convergence of $\{\lambda_t\}$ to the dual-optimal set $\Lambda^\star$ and primal feasibility:

$$\lim_{t\to\infty} \left(J_{C_k}(\theta_t) - \beta_k\right)_+ = 0, \qquad \lim_{t\to\infty} \text{dist}(\lambda_t, \Lambda^\star) = 0. \tag{12}$$

Finally, any limit point $(\theta^\star, \lambda^\star)$ satisfies the KKT conditions of the CMDP:

$$\nabla_\theta L(\theta^\star, \lambda^\star) = \mathbf{0}, \quad J_{C_k}(\theta^\star) \leq \beta_k, \quad \lambda_k^\star \geq 0, \quad \lambda_k^\star\left(J_{C_k}(\theta^\star) - \beta_k\right) = 0, \;\forall k. \tag{13}$$

## H.4 PROOF TEMPLATE MIRRORING A SMOOTHNESS STEP

For completeness, we restate the key inequality in the style of the referenced lemma. Let $F$ be as in Eq. 3, $g_t = \nabla F(\theta_t; \lambda)$, and define $\widehat{\theta}_{t+1} = \theta_t - \alpha_t g_t$ and $\theta_{t+1} = \theta_t - \alpha_t \widehat{g}_t$. By $L_\theta$-smoothness:

$$F(\theta_{t+1}) \leq F(\widehat{\theta}_{t+1}) + \left\langle \theta_{t+1} - \widehat{\theta}_{t+1}, \nabla F(\widehat{\theta}_{t+1}) \right\rangle + \tfrac{L_\theta}{2}\|\theta_{t+1} - \widehat{\theta}_{t+1}\|^2. \tag{14}$$

Taking expectations and using $\mathbb{E}[\widehat{g}_t \,|\, \mathcal{F}_t] = g_t + \varepsilon_t$,

$$\mathbb{E}[F(\theta_{t+1})] \leq \mathbb{E}\left[F(\widehat{\theta}_{t+1})\right] + \tfrac{L_\theta}{2}\alpha_t^2 \,\mathbb{E}\left[\|\widehat{g}_t - g_t\|^2\right]. \tag{15}$$

Also,

$$F(\widehat{\theta}_{t+1}) \leq F(\theta_t) + \left\langle \widehat{\theta}_{t+1} - \theta_t, \nabla F(\theta_t) \right\rangle + \tfrac{L_\theta}{2}\|\widehat{\theta}_{t+1} - \theta_t\|^2 = F(\theta_t) - \alpha_t\|g_t\|^2 + \tfrac{L_\theta}{2}\alpha_t^2\|g_t\|^2. \tag{16}$$

Combining Eqs. 15–16 and adding/subtracting $\varepsilon_t$ as in Eq. 9 gives

$$\mathbb{E}[F(\theta_{t+1})] - \mathbb{E}[F(\theta_t)] \leq -\alpha_t \,\mathbb{E}\left[\|g_t\|^2\right] + \tfrac{L_\theta}{2}\alpha_t^2 \,\mathbb{E}\left[\|g_t\|^2\right] + \tfrac{L_\theta}{2}\alpha_t^2 \,\mathbb{E}\left[\|\widehat{g}_t - g_t\|^2\right]. \tag{17}$$

which is the LC–MAPPO analogue of the inequality chain in Eqs. 14–17; the rest follows by summability of $\{\alpha_t^2\}$ and two–time–scale analysis for $\{\lambda_t\}$.

