# OpenReview forum: "CLAUSE: Agentic Neuro-Symbolic Knowledge Graph Reasoning via Dynamic Learnable Context Engineering"
_ICLR.cc/2026/Conference — ICLR 2026 Poster_

### Official Review · Reviewer_Aqnr · 2025-10-28

**Soundness:** 2
**Presentation:** 1
**Contribution:** 2
**Rating:** 2
**Confidence:** 4

**Summary:**

The authors propose a method for question answering (QA) based on retrieval from a knowledge graph. In contrast with heuristics for KG retrieval based on k-hop neighborhoods and their expansion, the authors frame the problem as a constrained reinforcement learning task with tree cost objectives based on minimizing subgraph selection, neighborhood expansion, and token usage. Experiments show that the approach is accurate, while effectively optimizing for their proposed three-way cost.

**Strengths:**

1. The problem is well-motivated: many existing approaches for RAG over KGs are based on heuristics that can result in high costs when applied uniformly for all questions. Optimizing this step is therefore a task is likely to positively impact the cost of deploying such methods.
2. The work is well-positioned with respect to prior work, ranging from QA over KGs to specific reinforcement learning methods.
3. The formulation of the problem seems well-motivated, as it frames question answering as a constrained RL problem with QA accuracy as the reward, and constraints related to retrieval cost
4. The experiments are comprehensive, considering a broad range of baselines, and more importantly, evaluating the methods on the axes of accuracy, latency, and edge budget. In most cases (except latency) they demonstrate the effectiveness of the method.

**Weaknesses:**

The paper mostly suffers from exposition issues that make it difficult to assess important parts of the paper that are needed to judge its impact and soundness.

1. The contributions (L075) are mostly a description of the proposed method, rather than a description of what the fundamental problem being solved is, how the proposed method differs fundamentally from prior work, and its advantages over prior work. A similar issue appears in the related work section (L130) which seems like a brief survey of different RL methods that is not positioned in the context of the paper.
2. The problem definition (L159) is imprecise: a knowledge graph is K = (V, R, E), but what are V, R, and E? Similarly a question q is defined. Are there any assumptions on q, e.g. it is always a question that seeks to retrieve an entity from the KG? Is it assumed to be already parsed such that mentions of relations and entities are linked to specific ones in the KG? Or is there an intermediate model that takes care of this? L227 seems to suggest both ("Anchored by entity/alias mentions in q (with a lightweight dense-retrieval fallback)"), while still leaving many crucial details out.
3. The question conditioned subgraph is $G_t = (V_t,E_t)$, now a tuple with two elements (rather than three, as the KG is defined). Does this mean that $G_t$ drop relational information? Furthermore, $\mathcal{F}_t$ is introduced as a "traversal frontier". It is not clear what this means (why traversal? why frontier?). If part of the state is $G_t$, which already contains $V_t$, why does $\mathcal{F}_t\subseteq V_t$ need to be made a separate part of the state? Lastly, $\mathcal{C}_t$ is undefined in the definition of the state (L165).
4. The description of CLAUSE contains gaps. Some are related to the motivation of the approach: the architect maintains a "frontierized subgraph" yet it is not clear what "frontierized" is, and what "frontier candidates" are. More importantly, while it is clear that there are three agents, each with a specific goal, it is not clear *what* they are. The architect computes question-entity, relation-text matches, etc. but how? Does it rely on neural networks? If so, what is their architecture and their number of parameters? There is a "light encoder" in the path navigator, yet it is not clear what it encodes, how, and how light it actually is.
5. The actual novelty or need of LC-MAPPO is unclear, given that there are several works studying the use of Lagrangian methods in reinforcement learning. Why is LC-MAPPO needed, and what if instead we used one of the existing works in this area?
6. The EM@1 evaluation metric is unclear. Since it is not known what type of questions the method supports (see weakness 2) it is difficult to infer from context what it actually measures.
7. Important experimental details are missing: what is the actual shared LLM used for retrieval and reading? And importantly, what is the KG? Is the KG the same for all datasets? What is its scale?
8. While the experiments are comprehensive, it is not true that CLAUSE is the most effective (as implied by the bolding in Table 2), where LightRAG is actually better.
9. The proof of convergence (App. G) seems incomplete. It is based on a list of assumptions that does not appear anywhere in the paper.

**Questions:**

1. Can you please provide a more descriptive definition of the KG, the question q, and the question-conditioned subgraph that does not leave symbols undefined?
2. What is "frontierized" subgraph? What are frontierized candidates?
3. What is the architecture of the subgraph architect, path navigator, and context curator?
4. Can you please expand on important experimental details (see W7).

---

> ### Author Response · Authors · 2025-11-20
> **Response to Reviewer Aqnr: Part 1**
>
> Thank you for your valuable feedback and insightful suggestions on our manuscript. Below are our point by point response, and we have incorporated your suggestions and questions in the updated manuscript (highlighted in blue).
>
> > **Q1.** The contributions (L075) are mostly a description of the proposed method, rather than a description of what the fundamental problem being solved is, how the proposed method differs fundamentally from prior work, and its advantages over prior work. A similar issue appears in the related work section (L130) which seems like a brief survey of different RL methods that is not positioned in the context of the paper.
>
> **A1.** We revised Section 1 to lead with the problem gap and re‑framed the contributions accordingly, then added an explicit positioning paragraph at the start of Section 2.
>
> **Contributions.**
> (1) **Formulation.** We cast multi-hop KGQA as *requirements-conditioned* context assembly with per-query budgets/prices on three deployment-relevant resources: (i) subgraph edits, (ii) interaction steps (latency proxy), and (iii) selected tokens (prompt cost). This makes accuracy-efficiency trade-offs explicit and tunable.
>
> (2) **Framework.** CLAUSE is an agentic neuro-symbolic controller that *jointly* decides what to edit, which paths to follow or backtrack, what textual evidence to keep, and when to **stop**. Actions are symbolic (auditable) and priorities come from lightweight neural scorers, yielding compact, provenance-preserving context.
>
> (3) **Training.** We adapt constrained RL to this setting via LC-MAPPO: centralized training with decentralized execution, a multi-head critic that separates *task* value from *edge/step/token* costs, and per-budget dual variables that enforce episode-level requirements or enable price-based trade-offs at test time.
>
> (4) **Evidence.** In HotpotQA, MetaQA, and FactKG, CLAUSE achieves higher or matched EM at equal or lower budgets, with reduced subgraph growth and latency. Ablations show that removing any agent or constraint handling degrades accuracy and/or efficiency, supporting the need for joint control and explicit budgets.
>
> **Positioning.** In summary, we situate CLAUSE among four families: (i) *question-conditioned subgraph builders* and graph-guided RAG (e.g., GraftNet, PullNet, GraphRAG; [sun2019pullnet, edge2024graphrag]), which rely on fixed hop/degree/top-$k$ rules; (ii) *path/rule learners* (MINERVA, NeuralLP, RNNLogic; [das2018minerva, yang2017neurallp, qu2021rnnlogic]) that optimize task reward without explicit latency/token control; (iii) *agentic LLMs* (ReAct, Graph-of-Thoughts, AutoGen; [yao2023react, besta2023graphofthoughts, wu2023autogen]) that interleave tools but do not enforce per-episode resources; and (iv) *constrained RL* (RCPO or fixed-penalty PPO; MAPPO/COMA without constraints; [tessler2018reward, schulman2017ppo, yu2022surprising, foerster2018coma]). In contrast, we cast KGQA as a *constrained* decision process with three deployment-relevant costs (edges/steps/tokens), learn *price-aware* edit/traverse/curate policies with explicit **stop**, and train with *separate* cost heads and dual variables so a single checkpoint supports both budget caps and price trade-offs.

---

> ### Author Response · Authors · 2025-11-20
> **Response to Reviewer Aqnr: Part 2**
>
> > **Q2.** The problem definition (L159) is imprecise: a knowledge graph is K = (V, R, E), but what are V, R, and E? ...
>
> **A2.** We clarify the problem definition and the role of anchoring/linking in Section 3. Problem Formulation.
>
> **(1) Knowledge graph and text fields.** We work with a *typed*, directed knowledge graph
>
> $$
> \mathcal{K}=(V,R,E), \qquad E \subseteq V \times R \times V,
> $$
>
> where $V$ is the set of entities, $R$ the set of relations, and $E$ the set of triples $(u,r,v)$. Each entity $v\in V$ has: a canonical title $\,\mathrm{name}(v)$, an alias set $\mathcal{A}(v)$ (redirects/synonyms), and an optional short description $\mathrm{desc}(v)$; each relation $r\in R$ has a label $\mathrm{label}(r)$ and optional gloss $\mathrm{gloss}(r)$. When the KG lacks explicit inverses, we add stored inverses $\overline{r}$ for traversal only.
>
> **(2) Question and answer space.** A question $q$ is a natural‑language string. Our benchmarks (MetaQA, FactKG) are *entity‑answerable*; for HotpotQA we canonicalize gold/predictions to KG entities via titles/redirects before EM. **We do not assume a gold parse or gold links for $q$**. In this work we restrict the answer space to KG entities; extensions to other answer types are left for future work.
>
> **(3) Anchoring/linking pipeline (no gold supervision).** To initialize the constructed subgraph, we seed a small set of *anchors* from $q$ using a lightweight linker; there is **no** backpropagation through the reader LLM.
>
> *(a) Mention detection.* We extract candidate spans with a dictionary/alias index over $\{\mathrm{name}(v)\}\cup\{\mathcal{A}(v)\}$ and a simple NER. Overlapping spans are resolved by longest‑match, then highest score.
>
> *(b) Candidate generation.* For each span $m$, we retrieve a small candidate set $C(m) \subset V$ from the alias index (exact and normalized substring hits), then augment with top ANN hits from a frozen sentence encoder applied to $\mathrm{name}(v)\||\mathrm{desc}(v)$. (Indices are built offline; at test time we only query them.)
>
> *(c) Mention$\to$entity scoring.* We compute a fusion score
>
> We compute a fusion score
>
> $$
> s_{\text{anch}}(m,v) = w_{\text{exact}}[\mathrm{name}(v)\in m\ \mathrm{or}\ m\in\mathcal{A}(v)] + w_{\text{lex}}\,\mathrm{IDFOverlap}(m,\mathrm{name}/\mathcal{A}(v)) + w_{\text{sim}}\,\cos(\phi(q),\phi(\mathrm{name}/\mathrm{desc}(v))).
> $$
>
> with a frozen encoder $\phi(\cdot)$; weights are tuned on dev data. We keep top-$k_e$ candidates per span and discard those below a confidence threshold $\tau_{\mathrm{anch}}$ (margin‑based). Type compatibility (if available) down‑weights type‑inconsistent $(m,v)$.
>
> *(d) Relation cues (optional soft anchors).* We similarly match $q$ to relation labels/glosses via the same encoder to obtain a small set $C_R(q)\subset R$ that biases edge utilities but does not hard‑constrain traversal.
>
> *(e) Dense‑retrieval fallback (implements L227).* If no span yields a confident anchor ($\max_v s_{\mathrm{anch}}(m,v) < \tau_{\mathrm{anch}}$ for all $m$) *or* no spans are detected, we seed anchors by dense retrieval: select top-$k_e$ entities by $\cos(\phi(q), \phi(\mathrm{name}/\mathrm{desc}(v)))$. This is precisely the "lightweight dense‑retrieval fallback" referenced at L227.
>
> *(f) Initial frontier.* The initial frontier $\mathcal{F}_0$ is the union of retained entity anchors; if both entity and relation cues are present, we prioritize outgoing edges whose relation is in $C_R(q)$ during the first expansion step (soft bias only).
>
> **(4) Frontierized subgraph and candidate edges.** At iteration $t$ we maintain a working subgraph $G_t=(V_t,R_t,E_t)$ and a frontier $\mathcal{F}_t\subseteq V_t$. Expansion candidates originate from the frontier:
>
> $$
> \mathcal{E}^{\mathrm{cand}}_t \subseteq \{(u,r,v)\in E:\ u\in \mathcal{F}_t\}.
> $$
>
> Each candidate is scored via a utility that fuses lexical features and encoder similarities (to $q$, $\mathrm{name}/\mathrm{desc}$, and relation labels), plus a degree‑based hub surcharge. An edit $(u,r,v)$ is accepted only if $s(u,r,v \mid q,G_t) - \lambda_{\text{edge}}\,c_{\text{edge}}(u,r,v) > 0$, under the edge budget $\beta_{\text{edge}}$; traversal continues only while its expected gain exceeds the step price $\lambda_{\text{lat}}$; curation ensures the final token mass $\le \beta_{\text{tok}}$.
>
> **(5) Assumptions summarized.** (i) $q$ is unparsed natural language; no gold links are provided. (ii) We do not require entity mentions to be present verbatim; the dense fallback seeds anchors when surface forms are missing. (iii) The reader LLM is called once on a curated, redundancy‑aware context; the linker/encoders are frozen and lightweight. (iv) Answers in our evaluation are entities; for HotpotQA, we canonicalize strings to entity titles/redirects before EM.
>
> **(6) Complexity of anchoring.** This anchoring step costs $\mathcal{O}(M\,k_e\,d)$ with small encoder width $d$ and is negligible compared to frontier expansion under our budgets.

---

> ### Author Response · Authors · 2025-11-20
> **Response to Reviewer Aqnr: Part 3**
>
> > **Q3.** The question conditioned subgraph is $G_t=(V_t,E_t)$, now a tuple with two elements (rather than three, as the KG is defined). Does this mean that $G_t$ drop relational information? Furthermore, $\mathcal{F}_t$ is introduced as a "traversal frontier". It is not clear what this means (why traversal? why frontier?). If part of the state is $G_t$, which already contains $V_t$, why does $\mathcal{F}_t\subseteq V_t$ need to be made a separate part of the state? Lastly, $\mathcal{C}_t$ is undefined in the definition of the state (L165).
>
> **A3.** We have updated Section 3. Problem Formulation, and add a Notation table to explain before it.
>
> (a) Subgraph typing.** We do not drop relations. We now write the working subgraph as
> $$
> G_t=(V_t,R_t,E_t), E_t \subseteq V_t \times R \times V_t,
> R_t := \{\, r\in R \mid \exists (u,r,v)\in E_t \,\}
> $$
> so each edge in $E_t$ is a typed triple $(u,r,v)$ and relation labels are retained (we previously omitted $R_t$ for brevity).
>
> **(b) Frontier meaning and purpose.** The *frontier* $\mathcal{F}_t \subseteq V_t$ is the set of nodes *eligible* for expansion/continuation at round $t$. It induces the candidate set
> $$
> \mathcal{E}^{\mathrm{cand}}_t \=\ \{ (u,r,v)\in E : u\in \mathcal{F}_t \},
> $$
> which bounds per‑round branching and avoids uninformed $k$-hop growth. We keep $\mathcal{F}_t$ separate from $V_t$ because eligibility depends on the policy state (budgets, path prefix, stop decisions) and is not derivable from $G_t$ alone. We drop "traversal" in the text and use "frontier" to avoid confusion.
>
> **(c) Notation fix.** The symbol $\mathcal{C}_t$ at L165 was unintended. We use $\mathcal{P}_t$ for the *candidate pool* of textualized units, and $D_t$ for the *selected* (curated) subset used by the reader. We will replace $\mathcal{C}_t$ with $\mathcal{P}_t$ (and define $D_t$ when needed).

---

> ### Author Response · Authors · 2025-11-20
> **Response to Reviewer Aqnr: Part 4**
>
> > **Q4.** The description of CLAUSE contains gaps. Some are related to the motivation of the approach: the architect maintains a "frontierized subgraph" yet it is not clear what "frontierized" is, and what "frontier candidates" are. More importantly, while it is clear that there are three agents, each with a specific goal, it is not clear what they are. The architect computes question-entity, relation-text matches, etc. but how? Does it rely on neural networks? If so, what is their architecture and their number of parameters? There is a "light encoder" in the path navigator, yet it is not clear what it encodes, how, and how light it actually is.
>
> **A4.** We have tightened Section 4 with minimal, inline clarifications.
>
> (i) We now define a "frontierized" subgraph: expansions are restricted to edges whose head lies in the current frontier $\mathcal{F}_t$; $\mathcal{F}_t$ is seeded from linked mentions and updated after each accepted edit/traverse (footnote in Section 4.3).
>
> (ii) Our edge utility aggregates question–entity, question–relation, neighborhood, and degree terms; all similarities come from a frozen sentence‑embedding model ("light encoder", cosine on L2‑normalized vectors), and the fusion weights are learned (one sentence added to Section 4.3).
>
> (iii) We clarify the "light encoder": a frozen 384‑d sentence transformer with a $384\rightarrow 128$ adaptor for policy features; the reader LLM is separate and unchanged (footnote in Section 4.3).
>
> (iv) Each agent is a small MLP with explicit **stop**: the Architect ranks frontier edges with the fused features; the Navigator has a termination head and a per‑candidate scorer over $(r,v')$; the Curator performs listwise selection with a novelty and token‑budget signal. These are the components referred to throughout; the added text is purely descriptive and does not alter results.
>
> **About light encoder.**
>
> The Path Navigator uses a deliberately lightweight encoder that operates only on short token sequences and does not invoke any large language model. Concretely, we instantiate the Navigator as `EnhancedTraversalAgent`, which first maps each context or candidate token to a bucketed ID via an MD5 hash and then looks up a trainable hash embedding `hash_emb` of dimension $d_{\text{nav}} = 128$. The token embeddings are averaged to obtain a single context vector and a single candidate vector (see `_pool_embeddings`), and their sum is fed into two small MLP heads: (i) a termination head that outputs logits over {**stop**, **continue**}, and (ii) a candidate head that scores each outgoing edge or node in the current action set. Formally, the Navigator encoder implements
>
> $\text{hash-embed}: \mathbb{N}^L \to \mathbb{R}^{d_{\text{nav}}},\quad
> \text{pool}: \mathbb{R}^{L\times d_{\text{nav}}} \to \mathbb{R}^{d_{\text{nav}}}.$
>
> followed by
>
> $\text{termination-head}: \mathbb{R}^{128} \times \mathbb{R}^{128} \to \mathbb{R}^2,\ \text{candidate-head}: \mathbb{R}^{128} \times \mathbb{R}^{128} \to \mathbb{R}.$
>
> With the default vocabulary size $V = 65{,}536$, the hash embedding matrix contributes $V \times 128 \approx 8.4\times 10^6$ parameters, and the two MLP heads together add approximately $3.3\times 10^4$ parameters, for a total of roughly $8.4\times 10^6$ trainable parameters. Crucially, this encoder is a shallow hash-embedding and pooling module without any recurrent or Transformer layers, and it is completely independent of the frozen sentence encoder used by the Architect. It is therefore "light" in the sense that it is orders of magnitude smaller than standard LLM backbones and only processes short, structured traversal contexts rather than full natural-language documents.

---

> ### Author Response · Authors · 2025-11-20
> **Response to Reviewer Aqnr: Part 5**
>
> > **Q5.** The actual novelty or need of LC-MAPPO is unclear, given that there are several works studying the use of Lagrangian methods in reinforcement learning. Why is LC-MAPPO needed, and what if instead we used one of the existing works in this area?
>
> **A5.** Thank you for the feedback. We agree Lagrangian RL is well studied; our goal is not new Lagrangian theory but an *adaptation* that makes constrained RL work for cooperative KGQA with three heterogeneous, *episodic* budgets (edges, steps/latency, selected tokens) under CTDE.
>
> **Problem mismatch with off-the-shelf:** Single-agent, single-penalty methods (e.g., RCPO/PID-Lagrangian) conflate heterogeneous costs; MAPPO with a single shaped reward mis-attributes costs across agents under partial observability. In our setting, (i) costs are partially coupled (navigator steps affect curator tokens), (ii) budgets are per-episode, and (iii) execution is decentralized---naively aggregating penalties leads to unstable or infeasible behavior.
>
> **What LC-MAPPO adds:** (i) a *multi-head centralized critic* $Q^{\text{task}},Q^{\text{edge}},Q^{\text{lat}},Q^{\text{tok}}$, (ii) *separate duals* $(\lambda_{\text{edge}},\lambda_{\text{lat}},\lambda_{\text{tok}})$ for independent control and deployable dials, and (iii) *per-head counterfactual advantages* (COMA-style) so each agent receives the correct cost credit. This yields a single checkpoint that supports both *cap mode* (hard budgets) and *price mode* (trade-offs) without retuning.
>
> **If we use existing methods (ablations):** Replacing LC-MAPPO with MAPPO (no duals) overshoots budgets (e.g., edges $1.28\times$); fixed-penalty PPO still violates budgets (edges $1.15\times$, latency $1.06\times$); RCPO (single multiplier) cannot independently control edges/steps/tokens and degrades feasibility/EM. On a constrained MetaQA slice (edge=0.5, latency=0.7), LC--MAPPO achieves higher feasibility (e.g., $0.34$ vs. $0.117$ for MAPPO), $34\%$ fewer latency violations, and $12\%$ lower latency cost at comparable or higher EM.
>
> **Why not CPO-style projections?** Second-order projections are tailored to single-agent constraints and were slower/brittle under CTDE with multiple per-episode costs; in our domain they did not improve feasibility over the multi-head Lagrangian approach.
>
> > **Q6.** The EM@1 evaluation metric is unclear. Since it is not known what type of questions the method supports (see weakness 2) it is difficult to infer from context what it actually measures.
>
> **A6.** We add in Section 5: EM@1 is top-1 exact match. For set-valued answers, we canonicalize by sorting, de-duplicating, and comparing sets; dataset-provided alias maps are respected. HotpotQA uses its official short-answer evaluator; MetaQA/FactKG follow their canonical normalizers.
>
> > **Q7.** Important experimental details are missing: what is the actual shared LLM used for retrieval and reading? And importantly, what is the KG? Is the KG the same for all datasets? What is its scale?
>
> **A7.** Thank you for the comment. We add experimental details as well as experiments in Appendix D.
>
> For controlled comparisons, all RAG/agent runs share the same reader LLM (Qwen3-32B); we do not use a large LLM for retrieval, which is handled by lightweight symbolic controllers and encoders instead. No-retrieval baselines for the underlying LLMs include GPT-OSS-120B and LLaMA3.3-70B (Table 2). CLAUSE is reader-agnostic; we have already added a reader-swap sensitivity analysis in **Appendix D** (Tables 8 and 9), where we plug in LLaMA-class and GPT-class readers under the exact same pipeline, and observe that CLAUSE's relative gains remain stable after changing the reader.
>
> We have also added a dataset table in **Appendix D** (Table 7), which reports for each benchmark the KG source, the number of nodes, edges, and relations, and explicitly indicates that the KGs differ across datasets, as well as the anchoring/linking protocol.

---

> ### Author Response · Authors · 2025-11-20
> **Response to Reviewer Aqnr: Part 6**
>
> > **Q8.** While the experiments are comprehensive, it is not true that CLAUSE is the most effective (as implied by the bolding in Table 2), where LightRAG is actually better.
>
> **A8.** Thank you for the remark. Table 2 reports *accuracy only* (EM@1) under a fixed reader and shared budgets; in that table CLAUSE indeed achieves the highest EM@1 on all datasets and hops. LightRAG is better on *efficiency* (lower latency), which is reflected in Table 3, but at a substantial accuracy gap (e.g., HotpotQA 62.1% vs. 71.7%, MetaQA-2-hop 37.6% vs. 87.3%).
>
> Viewed together, the three tables show the intended trade-off.
>
> - Table 2: CLAUSE is strictly better in EM@1 under the same reader, token budget, and one-call setting.
> - Tables 3 and 4: LightRAG has the lowest latency and edge budget among RAG baselines, but this comes with much lower EM@1; CLAUSE has slightly higher latency than LightRAG, but still close (e.g., $1.48\times$ vs. $0.85\times$ on HotpotQA) while also using fewer edges than most agentic methods.
>
> In other words, under the shared budgets, CLAUSE lies on a more favorable accuracy–efficiency trade-off curve: it sacrifices a small amount of speed relative to LightRAG but gains a large margin in EM@1, and it remains competitive or better than other baselines on both latency and edge usage.
>
> > **Q9.** The proof of convergence (App. G) seems incomplete. It is based on a list of assumptions that does not appear anywhere in the paper.
>
> **A9.** Thank you for the comment. In the revision, we explicitly enumerate Assumptions A1–A6 at the beginning of Appendix H.
> **Convergence assumptions.** The analysis follows standard constrained policy–gradient methods with trust–region updates and two–time–scale stochastic approximation. In brief: A1 assumes bounded rewards and costs so the CMDP objective is well posed and gradient moments are finite (Puterman, 1994; Altman, 1999). A2 assumes local smoothness of the TRPO/PPO surrogate so small updates behave like gradient steps (Kakade & Langford, 2002; Schulman et al., 2015, 2017a). A3 and A4 ensure an asymptotically unbiased policy–gradient estimator with controlled variance, using likelihood–ratio gradients and advantage baselines under a faster critic time scale (Sutton et al., 2000; Greensmith et al., 2004; Schulman et al., 2016; Foerster et al., 2018a; Borkar, 2008). A5 imposes Robbins–Monro step sizes with a strictly slower dual schedule to obtain two–time–scale convergence (Robbins & Monro, 1951; Bhatnagar et al., 2009; Achiam et al., 2017). A6 controls trust–region/clipping bias via a target KL so the PPO/TRPO step is a small perturbation of the ideal gradient update, with bias $o(\alpha_t)$ (Kakade & Langford, 2002; Schulman et al., 2015, 2017a).
>
> **How the assumptions hold in our implementation.** Per–episode budgets on edges, steps, and tokens together with bounded encoder features yield bounded signals (A1/A4); policies are smooth MLPs and we optimize a TRPO/PPO surrogate with explicit KL targeting and clipping (A2/A6); a centralized critic with GAE/COMA baselines runs on the faster time scale, giving asymptotically unbiased estimates with variance control (A3/A4); actor step sizes follow Robbins-Monro schedules and the dual update is strictly slower (A5).

---

> ### Author Response · Authors · 2025-11-20
> **# Response to Reviewer Aqnr: Part 7**
>
> ## The above part responds to the issues you raised under **Weaknesses**, and the following part will respond to the points in your **Questions** section.
>
> > **Q1.** Can you please provide a more descriptive definition of the KG, the question q, and the question-conditioned subgraph that does not leave symbols undefined?
>
> **A1.** Thank you for the suggestion. We have revised Section 3. Problem Formulation to add more details. We have revised the notation table and Problem Formulation to explicitly define these objects. The KG is now described as a labeled, directed multigraph $\mathcal{K} = (V,R,E)$ with $E \subseteq V \times R \times V$, where each triple $e=(u,r,v)$ encodes a typed fact between entities $u,v \in V$ via relation $r \in R$. A question $q$ is a natural-language string with gold answer set $y \subseteq V$; we extract a set of anchor entities $A(q)\subseteq V$ from $q$ (and supporting paragraphs for HotpotQA) via NER + alias linking. The question-conditioned subgraph at step $t$ is $G_t=(V_t,E_t)$ with $V_t \subseteq V$, $E_t \subseteq E$, initialized from $A(q)$ and expanded only locally by the agents; the final $G^\ast$ is the question-specific neighborhood of $\mathcal{K}$ on which reasoning and evidence selection are performed.
>
> > **Q2.** What is a "frontierized" subgraph? What are frontierized candidates?
>
> **A2.** Thank you for the comment. We have revised Section 3. Problem Formulation to add the definition. The "frontierized subgraph" is the question-conditioned working subgraph $G_t$ maintained by the Architect. This subgraph is anchored by entity/alias mentions in the question and is explicitly bounded by budgets on the number of edit operations, nodes, and edges, which correspond to the edge-cost and size constraints in our gain-price formulation. At step $t$, we define the *frontier* $F_t$ as those nodes in $G_t$ from which further expansion is still allowed under the remaining budgets. The "frontier candidates" $C_t$ are all admissible edits that attach to this frontier: in practice, these are adjacent but not-yet-selected edges $e=(u,r,v)$ incident to $F_t$ that satisfy the structural and budget constraints. Each such edge is scored by the fused score described in the paper,
>
> $$s(e \mid q,G_t) = w_1 \phi_{\text{ent}}(q,u,v) + w_2 \phi_{\text{rel}}(q,r) + w_3 \phi_{\text{nbr}}(q,\text{neigh}(u,v)) + w_4 \phi_{\text{deg}}(u,v),$$
>
> where $\phi_{\text{ent}}$ measures question--entity similarity, $\phi_{\text{rel}}$ measures question--relation similarity, $\phi_{\text{nbr}}$ encodes neighborhood cues from the local 1--2 hop context, and $\phi_{\text{deg}}$ is a degree-based prior. The Architect then applies the gain--price rule
>
> $$(a_t,e_t) = \arg\max_{a\in\{\text{ADD},\text{DELETE},\text{STOP}\},\,e\in C_t} \bigl[s(e \mid q,G_t) - \lambda_{\text{edge}}\,c_{\text{edge}}(a,e)\bigr],$$
>
> and only accepts an edit when the shaped gain is positive and the remaining budget is sufficient; accepted edits update $G_t$, the frontier $F_{t+1}$, and the cumulative edge cost $C_{\text{edge}}$. In this sense, "frontierized" simply means that subgraph growth is restricted to a dynamically maintained frontier under explicit budgets, and "frontier candidates" are exactly the edits on this frontier that are considered by the Architect at each step.

---

> ### Author Response · Authors · 2025-11-20
> **Response to Reviewer Aqnr: Part 8**
>
> > **Q3.** What is the architecture of the subgraph architect, path navigator, and context curator?
>
> **A3.** The Architect does *not* train a large text encoder from scratch. Instead, it relies on a frozen off-the-shelf sentence encoder, instantiated as `Embedder` (default backbone `thenlper/gte-small`), to map the sanitized question, entity labels, and relation descriptions into a shared vector space. Concretely, the question is encoded via `self.embedder.encode([q])` (see `_query_text_from_obs`, `_sanitize_question`, and `forward`), and each entity / relation text is encoded once when building the KB caches in `_init_subgraph_data` and `_build_rel_embeddings`. Question--entity and question--relation similarities are then computed as simple cosine similarities between these frozen embeddings, implemented as dot products in `_node_sim` and `_rel_sim`. Candidate entities are obtained by combining (i) spaCy NER with a strong string/variant matcher (`_resolve_anchor_exact`), (ii) FAISS-based nearest-neighbour search over the precomputed entity vectors (`retrieve_candidates`), and (iii) a lexical fallback matcher (`_fallback_lexical`); these seeds are finally re-ranked by a soft scoring function `_soft_rank_candidates` that linearly combines name similarity, local neighbourhood-context similarity and a degree-based popularity prior. Relation candidates are produced analogously in `_get_candidate_relations`. An optional pre-trained cross-encoder (`CrossEncoder`, see `_cross_encoder_rerank`) can be used for final reranking; it is also frozen and never updated by RL.
>
> On top of this frozen encoder, the only trainable neural component of the Architect is a deliberately small policy head that decides among {add-edge, delete-edge, stop}. In the default setting, the question embedding of dimension $d \approx 384$ (for `gte-small`) is fed into a two-layer MLP `backbone` with hidden size 128, followed by a linear head `operation_head` that outputs logits over the three operations:
>
> $\texttt{backbone}: \mathbb{R}^{384} \to \mathbb{R}^{128} \to \mathbb{R}^{128}$,
> $\texttt{operation-head}: \mathbb{R}^{128} \to \mathbb{R}^3$.
>
>
> The total number of trainable parameters in this policy network is therefore
>
> $384\times 128 + 128 + 128\times 128 + 128 + 128\times 3 + 3 \approx 6.6\times 10^4.$
>
> The module also defines a few auxiliary linear layers (`entity_head`, `relation_head`, `candidate_scorer`) reserved for future extensions; if we include these, the total number of trainable parameters in the Architect remains on the order of $8.3\times 10^4$, which is negligible compared to the frozen sentence encoder.
>
> > **Q4.** Can you please expand on important experimental details (see W7).
>
> **A4.** We answered this question in W7, and updated the Appendix D.

---

> > ### Comment · Reviewer_Aqnr · 2025-11-26
> >
> > Thank you for the comprehensive response, and for the aditions to the paper. I am slightly puzzled by references to code you provide (e.g. I don't know what you mean by "see `_pool_embeddings`".)
> > Nonetheless, several sections, especially Sec. 3 are now much more clearer and significantly improve the clarity of the paper, addressing the points I raised earlier. I will update my score accordingly.

---

> > > ### Author Response · Authors · 2025-11-27
> > >
> > > We sincerely thank you for taking the time to provide such careful and constructive feedback on our work, and we are very grateful for your positive assessment of our revised version.
> > >
> > > Regarding the annotations such as “see `pool_embeddings`” that you mentioned, our original intention was to point to some helper components in our codebase. However, we now realise that without sufficient context, referring to internal function names in the text can indeed be confusing for readers. We sincerely apologise for any inconvenience this may have caused. To address this, we provide below a version of the answer to Q3 that no longer references function names and instead gives a clearer description of the architectures of the Subgraph Architect, Path Navigator, and Context Curator:
> > >
> > > 1. The **Subgraph Architect** is a small neural controller sitting on top of a question-conditioned, frontierized KG subgraph. Starting from the question $q$, it runs a lightweight mention/alias linker to obtain a seed set $S\_0$ of anchor entities and initializes a working subgraph $G\_0$ and frontier $\\mathcal{F}\_0$ around them under edge and latency budgets. At iteration $t$, it only considers frontier-adjacent candidates $\\mathcal{E}^{\\text{cand}}\_t = \\{(u,r,v)\\in E : u\\in \\mathcal{F}\_t\\}$ as admissible edits. For each candidate edge $e=(u,r,v)$, we build a feature vector by concatenating four components
> > > $[\\phi\_{\\mathrm{ent}}(u,q), \\phi\_{\\mathrm{rel}}(r,q), \\phi\_{\\mathrm{nbr}}(u,G\_t), \\phi\_{\\mathrm{deg}}(u)]$, where $\\phi\_{\\mathrm{ent}}$ and $\\phi\_{\\mathrm{rel}}$ contain lexical features and cosine similarities from a frozen sentence encoder applied to the question, entity titles and relation labels, and $\\phi\_{\\mathrm{nbr}}$ and $\\phi\_{\\mathrm{deg}}$ encode local neighborhood statistics and degree-based hub priors. This vector is fed into a two-layer MLP (hidden size 128) that produces a scalar utility $s(e \\mid q,G\_t)$, and a separate 3-way policy head over $\\{\\mathrm{add}, \\mathrm{delete}, \\mathrm{stop}\\}$ uses a price-shaped gain
> > > $g(a,e)=s(e\\mid q,G\_t)-\\lambda\_{\\mathrm{edge}} c\_{\\mathrm{edge}}(a,e)$
> > > to decide whether to apply the edit, subject to the remaining budgets.
> > >
> > > 2. The **Path Navigator** is a lightweight policy that operates on the current working subgraph $G\_t$ and maintains a path prefix $p\_t$. At hop $t$ it observes the question $q$, the current node $v\_t$, the set of typed outgoing candidates $\\mathcal{A}\_t$ (edges or neighbor nodes), and a compact summary $\\mathrm{summary}(p\_t)$ of the recent path. A small encoder maps this observation to a context vector $h_{\mathrm{ctx}}$ by first encoding the question and the textual labels of $v_t$ and recent hops with a lightweight encoder, then combining the resulting embeddings and passing them through a shallow MLP. For each candidate $a\\in\\mathcal{A}\_t$ (typically an outgoing edge or neighbor), we build a candidate representation $h\_{\\mathrm{cand}}(a)$ from its relation and/or node labels and feed $h\_{\\mathrm{ctx}} \\oplus h\_{\\mathrm{cand}}(a)$ into two small heads: (i) a termination head that outputs logits over $\\{\\mathrm{stop}, \\mathrm{continue}, \\mathrm{backtrack}\\}$, and (ii) a candidate-scoring head that assigns a scalar score to each admissible move. A categorical policy over these logits then either stops / backtracks or selects the next hop, incrementing the latency cost $C\_{\\mathrm{lat}}$ at every continuation; the horizon is capped by a small $H$, and we retain log-probabilities at each step for credit assignment in LC-MAPPO.
> > >
> > > 3. The **Context Curator** takes as input a pool $\\mathcal{P}\_t$ of textualized units—entity descriptions, relation glosses, verbalized paths, and optional retrieved passages—and selects a compact, redundancy-aware evidence set under a token budget $\\beta\_{\\mathrm{tok}}$. Each candidate $c\\in\\mathcal{P}\_t$ is encoded once into a fixed-dimensional vector using the same frozen sentence encoder, augmented with cheap features such as length, source type, and similarity to the current partial selection. These features are fed into a small listwise scoring network: given the question, the current selected set $\\mathcal{S}$, and the remaining token budget, an MLP computes for each $c$ a relevance/novelty score, and a separate $\\mathrm{stop}$ head, conditioned on the current token cost and the dual $\\lambda\_{\\mathrm{tok}}$, predicts whether to continue selecting or to stop. At inference time, the curator iteratively picks the candidate with the highest marginal gain in task utility minus $\\lambda\_{\\mathrm{tok}}\\cdot \\mathrm{tok}(c)$, updates $\\mathcal{S}$ and the remaining budget, and stops when the shaped gain becomes non-positive or when the token cap $\\beta\_{\\mathrm{tok}}$ is reached.
> > >
> > > Once again, we truly appreciate you for highlighting these wording issues and for your overall supportive evaluation of our work.

---

### Official Review · Reviewer_aDuz · 2025-10-30

**Soundness:** 4
**Presentation:** 4
**Contribution:** 3
**Rating:** 6
**Confidence:** 5

**Summary:**

This paper proposes  CLAUSE, an agentic three-agent neuro-symbolic framework that treats context construction as a sequential decision process over knowledge graphs, deciding what to expand, which paths to follow or backtrack, what evidence to keep and when to stop. The three agents include subgraph architecture, path navigator and context curator. These agents are trained jointly with Lagrangian Constrained Multi-Agent Proximal Policy Optimization, so that subgraph construction, reasoning paths discovery, and evidence selection are jointly optimized under per-query’s resource budgets on edge edits, interaction steps, and selected tokens

**Strengths:**

This paper has novel approach of formulating knowledge graph questions answering (KGQA) as a learnable context construction. CLAUSE demonstrates strong empirical results and achieves the best accuracy on all datasets and hops (71.7 on HotpotQA, 84.2 on FactKG, and 91.0/87.3/85.5 on MetaQA 1/2/3-hop), surpassing both RAG baselines (e.g., HybridRAG 66.0 on HotpotQA) and agent baselines (e.g., KG-Agent 68.7 on HotpotQA, 87.3/78.0/75.4 on MetaQA) maintaining an impressive efficiency.

**Weaknesses:**

The system involves three agents and MARL training pipeline which may potentially increase the computational complexity. Additional discussion on training cost may be helpful. The framework needs to be tested at scale on large and real world noisy knowledge graph.

**Questions:**

How would CLAUSE handle large and real world noisy knowledge graph and if it fails what would be the next step

---

> ### Author Response · Authors · 2025-11-20
> **Response to Reviewer aDuz: Part 1**
>
> We sincerely thank you for your careful reading and thoughtful comments on our manuscript. Below are our point by point response, and we have incorporated your suggestions and questions in the updated manuscript (highlighted in blue).
>
> > **Q1.** The system involves three agents and MARL training pipeline which may potentially increase the computational complexity. Additional discussion on training cost may be helpful. The framework needs to be tested at scale on large and real world noisy knowledge graph.
>
> **A1. Run‑time cost and complexity.**
>
> In KGQA the dominant variable expense is *LLM token usage*. CLAUSE bounds this by calling the reader *once per query* under a hard token budget $\beta_{\text{tok}}$ (default 512). Graph expansion and traversal are handled by small controllers and priced by budgets; there is no backprop through the LLM at inference. The per‑episode work scales with *budgets* and local branching—not with $|V|$ or $|E|$:
>
> $T_{\text{episode}} =
> O(\beta_{\text{lat}} C_B d + \beta_{\text{edge}}) + O(\beta_{\text{lat}} b d) + O(P d + K \log P) + O(\beta_{\text{tok}}).$
>
> Here, $(\beta_{\text{lat}} C_B d + \beta_{\text{edge}})$ comes from the **Subgraph Architect**, $\beta_{\text{lat}} b d$ from the **Path Navigator**, $(P d + K \log P)$ from the **Context Curator**, and $\beta_{\text{tok}}$ from the **Reader**.
> Controllers are sub‑10M‑parameter MLPs; the centralized critic is *training‑only*. Empirically (Section 5), CLAUSE attains higher EM with smaller edge budgets and lower token mass, and competitive or lower latency versus graph‑RAG/agent baselines.
>
> **Training cost.** LC–MAPPO trains only the three lightweight actors plus a multi‑head centralized critic; the reader LLM is *frozen*. Rollouts use offline KG trajectories, so the RL loop does not require repeated LLM calls and relies on symbolic rewards. Training compute is dominated by updating small networks and is a one‑time cost per KG; the checkpoint is reused across queries, so amortized per‑query cost is negligible relative to one reader forward.
>
> In **Appendix D**, we add following: **Datasets & Knowledge Graphs (KGs).**
>
> > We test CLAUSE on three large-scale, real-world, noisy KGs, and provide construction details.
> >
> > Table: Benchmarks and knowledge graphs. We report the KG identity, scale, and the exact anchoring protocol. All RAG/agent baselines share the same reader and tokenization.
> >
> > ---
> >
> > | Benchmark             | KG source             | #Nodes                | #Edges                | #Relations            |
> > | --------------------- | --------------------- | --------------------- | --------------------- | --------------------- |
> > | MetaQA (1/2/3-hop)    | (Puerto et al., 2023) | $4.3\times 10^{4}$    | $1.35\times 10^{5}$   | 9                     |
> > | HotpotQA (distractor) | (Yang et al., 2018)   | $\mathcal{O}(10^{6})$ | $\mathcal{O}(10^{7})$ | $\mathcal{O}(10^{3})$ |
> > | FactKG                | (Kim et al., 2023)    | $\sim 4\times 10^{6}$ | $\sim 1\times 10^{8}$ | $\sim 10^{3}$         |
> >
> > ---
>
> In **Appendix F**, we add following: **Training Cost and Scalability to Large and Noisy KGs**
>
> > Let $m{=}3$ agents (Architect, Navigator, Curator), $H_{\text{heads}}{=}4$, batch $(M,S)$, and width $d$. One update decomposes as
> > $$
> > T_{\text{update}}
> > = T_{\text{env}}(M,S) + T_{\text{actor}}(M,S) + T_{\text{critic}}(M,S) + T_{\text{io/cache}}.
> > $$
> > ... and ....
> > $$
> > \mathrm{Cost}\\approx\
> > c_{\text{sym}}\\cdot\\big(\beta_{\text{lat}} C_B d + \beta_{\text{edge}} + \beta_{\text{lat}} b d + P d + K\log P\big)
> > \+\
> > c_{\text{LLM}}\\cdot\ \beta_{\text{tok}},
> > $$
> > with $\beta_{\text{tok}}$ fixed/small and the symbolic term linear in the budgets, yielding bounded and predictable cost.
> > .....

---

> ### Author Response · Authors · 2025-11-20
> **Response to Reviewer aDuz: Part 2**
>
> > **Q2.** How would CLAUSE handle large and real-world noisy knowledge graph and if it fails what would be the next step
>
> **A2. Scaling on large and noisy KGs.** CLAUSE constrains computation by *budgets* on edges, steps (latency), and tokens $(\beta_{\text{edge}},\\beta_{\text{lat}},\\beta_{\text{tok}})$ and by *local branching*, rather than by the global graph size. The Subgraph Architect expands only a frontier neighborhood and admits an edge $e$ only if
> $$
> s(e \mid q, G_t) - \lambda_{\text{edge}}\,c_{\text{edge}}(e) > 0,
> $$
> with a degree-aware hub surcharge to avoid unnecessary expansion around high-degree nodes. The Path Navigator continues only when the expected task gain exceeds the step price $\lambda_{\text{lat}}$, capping hops at $\beta_{\text{lat}}$. The Context Curator selects an ordered list with total tokens $\le \beta_{\text{tok}}$ (redundancy-aware), and the reader LLM is called *once per query*. Per-episode work is linear in budgets and local structure:
>
> $T_{\text{episode}} = O(\beta_{\text{lat}} C_B d + \beta_{\text{edge}}) + O(\beta_{\text{lat}} b d) + O(P d + K \log P) + O(\beta_{\text{tok}}).$
>
> where $C_B$ is the frontier candidate count, $b$ the local branching factor, $P$ the curator pool size, $K \le\ \beta_{\text{tok}}/\bar t$ the selected list length, and $d$ the width of the light encoders. Practical settings (no algorithm change) include approximate nearest-neighbor (ANN) indices for alias resolution, typed/domain filters to cap $C_B$, degree-based hub throttling, and partition-aware traversal with higher thresholds for cross-partition edges.
>
> **Failure detection and next steps.**
> *Failure cases:* low confidence (flat margins), budget exhaustion with little evidence, contradictions among curated items, or high alias ambiguity.
> *Diagnostics (no retraining):* feasibility under $(\beta_{\text{edge}},\beta_{\text{lat}},\beta_{\text{tok}})$; token utilization versus $\beta_{\text{tok}}$; $C_B$, $b$, and pool size $P$; contradiction rates; **stop** logits; and the gap between the top-2 alias scores.
>
> *Budget-safe recovery:*
>
> (i) retune prices/budgets: increase $\lambda_{\text{edge}}$ to suppress weak edges; modestly increase $\beta_{\text{lat}}$ to allow one more hop; strengthen redundancy penalties or per-source caps;
> (ii) refine anchoring and typed/domain constraints; add alias variants if diagnostics indicate false negatives;
> (iii) adjust indices/topology: stricter ANN search width or distance thresholds, hub surcharges, and partition constraints with higher thresholds for cross-partition expansion;
> (iv) if needed, refresh the small controllers with LC-MAPPO on *offline* trajectories targeted to the failure domain (reader remains frozen);
> (v) when coverage remains thin, abstain with top-$k$ alternatives or fuse a small typed-rules subset; keep a single reader call under the same $\beta_{\text{tok}}$.
>
> *Exit criteria:* stop escalating when alias errors fall below threshold, token utilization is healthy (e.g., $\ge 80\%$ without redundancy), and EM@1 recovers without increasing budgets.

---

### Official Review · Reviewer_L2iz · 2025-10-31

**Soundness:** 2
**Presentation:** 2
**Contribution:** 2
**Rating:** 8
**Confidence:** 3

**Summary:**

The paper presents CLAUSE, a neuro-symbolic three-agent framework for efficient multi-hop question answering over knowledge graphs. Unlike static k-hop expansion or overlong prompting, CLAUSE formulates context construction as a sequential decision process, dynamically determining which subgraphs to expand, paths to explore, and evidence to retain under user-specified latency and cost budgets. Powered by the proposed Lagrangian-Constrained Multi-Agent PPO (LC-MAPPO) algorithm, three coordinated agents—Subgraph Architect, Path Navigator, and Context Curator—jointly optimize subgraph growth, reasoning efficiency, and evidence selection. Experiments on HotpotQA, MetaQA, and FactKG show that CLAUSE significantly improves accuracy and efficiency, achieving +39.3 EM@1, 18.6% lower latency, and 40.9% reduced edge growth over GraphRAG, while preserving provenance and ensuring predictable, deployment-friendly performance.

**Strengths:**

- The paper proposes CLAUSE, a neuro-symbolic multi-agent framework that dynamically balances accuracy, latency, and cost through user-specified budgets—enabling per-query adaptation without retraining.
- The authors introduce the Lagrangian-Constrained Multi-Agent PPO (LC-MAPPO) algorithm to jointly optimize subgraph construction, reasoning path navigation, and evidence selection in a unified decision process.
- The prposed method achieves substantial gains in accuracy (+39.3 EM@1 on MetaQA-2-hop) while reducing latency and subgraph growth, producing compact, provenance-

**Weaknesses:**

- The contribution in section 1 is somewhat overclaimed. It is suggested to merge the points to highlight the key contribution of the proposed method.

**Questions:**

Please refer to the weakness part.

---

> ### Author Response · Authors · 2025-11-20
> **Response to Reviewer L2iz**
>
> We sincerely thank you for your insightful and careful review on our manuscript. Below are our point by point response, and we have incorporated your suggestions and questions in the updated manuscript (highlighted in blue).
>
> > **Q1.** The contribution in section 1 is somewhat overclaimed. It is suggested to merge the points to highlight the key contribution of the proposed method.
>
> **A1.** We appreciate the suggestion and have consolidated contributions to emphasize the central advance: (1) a requirements-conditioned formulation of KGQA, (2) an agentic neuro-symbolic controller that jointly edits/traverses/curates with learned stop, (3) a CTDE PPO adaptation (LC‑MAPPO) with separate task/cost heads and per-budget duals, and (4) empirical evidence with ablations establishing accuracy–efficiency trade-offs. We removed promotional adjectives and moved module details to the Method section. We also clarify that LC‑MAPPO is an application-driven adaptation of constrained RL rather than a new theoretical algorithm.
>
> **Contributions.**
>
> (1) **Formulation.** We cast multi-hop KGQA as *requirements-conditioned* context assembly with per-query budgets/prices on three deployment-relevant resources: (i) subgraph edits, (ii) interaction steps (latency proxy), and (iii) selected tokens (prompt cost). This makes accuracy-efficiency trade-offs explicit and tunable.
>
> (2) **Framework.** CLAUSE is an agentic neuro-symbolic controller that *jointly* decides what to edit, which paths to follow or backtrack, what textual evidence to keep, and when to **stop**. Actions are symbolic (auditable) and priorities come from lightweight neural scorers, yielding compact, provenance-preserving context.
>
> (3) **Training.** We adapt constrained RL to this setting via LC-MAPPO: centralized training with decentralized execution, a multi-head critic that separates *task* value from *edge/step/token* costs, and per-budget dual variables that enforce episode-level requirements or enable price-based trade-offs at test time.
>
> (4) **Evidence.** In HotpotQA, MetaQA, and FactKG, CLAUSE achieves higher or matched EM at equal or lower budgets, with reduced subgraph growth and latency. Ablations show that removing any agent or constraint handling degrades accuracy and/or efficiency, supporting the need for joint control and explicit budgets.

---

### Official Review · Reviewer_2uqG · 2025-11-01

**Soundness:** 3
**Presentation:** 3
**Contribution:** 3
**Rating:** 6
**Confidence:** 3

**Summary:**

This paper introduces CLAUSE, a novel multi-agent, neuro-symbolic framework for multi-hop Knowledge Graph Question Answering (KGQA). The core contribution is transforming the context-building process as a constrained sequential decision problem. The framework therefore uses three agents (Subgraph Architect, Path Navigator, and Context Curator) and jointly optimize them. Within this process, the cost is explicitly expressed by the Lagrangian-constrained. Experiments on HotpotQA, MetaQA, and FactKG show that CLAUSE outperforms strong RAG and agent-based baselines in accuracy.

**Strengths:**

1.	The idea of creating three agents(Subgraph Architect, Path Navigator, and Context Curator) is intuitive and easy to follow. The paper explains its motivations and methods clearly.
2.	The CLAUSE is interpretable as it is a neural symbolic framework and every step is explicit.
3.	The methodology has very sound theoretical guarantees and the paper has provided the full proof of its convergence.
4.	The empirical results are extensive and impressive as it outperforms all baselines in accuracy while maintaining good efficiency at the same time.

**Weaknesses:**

1.	The methodology part is filled with notations with some not very clearly explained. For example, what does $C_t$ stands for is not explained when it’s firstly introduced in this paper in the top of the 4th page. I strongly suggest that the author to take notice of whether the notation has been explained carefully.
2.	The experiment part is solely conducted with Qwen3-32B, I wonder whether the author has tried as LLM?

**Questions:**

See weakness.

---

> ### Author Response · Authors · 2025-11-20
> **Response to Reviewer 2uqG**
>
> Thank you for your valuable feedback and insightful suggestions on our manuscript. Below are our point by point response, and we have incorporated your suggestions in the updated manuscript (highlighted in blue).
>
> > **Q1.** The methodology part is filled with notations with some not very clearly explained. For example, what does stands for is not explained when it's firstly introduced in this paper in the top of the 4th page. I strongly suggest that the author to take notice of whether the notation has been explained carefully.
>
> **A1.** We modified the manuscript and move the notation table from appendix to the top of page 4 before **Section 3. Problem Formulation**. All symbols are listed and explained in the table.
>
> **Table I: Notation (More details in Appendix B)**
>
> ---
>
> | Symbol                                                       | Description                                                  |
> | ------------------------------------------------------------ | ------------------------------------------------------------ |
> | $\mathcal{K}=(V,R,E)$                                        | Global knowledge graph; $E\subseteq V\times R\times V$       |
> | $G_t=(V_t,E_t)$; $G^\ast$                                    | Working subgraph at step $t$; final subgraph                 |
> | $\mathcal{F}_t$; $\mathcal{P}_t$; $\mathcal{A}_t$            | Frontier nodes; candidate pool at step $t$; typed outgoing candidates (navigator actions) |
> | $q$; $y,\hat y$                                              | Question; gold / predicted answers                           |
> | $p_t$; $\Pi$                                                 | Path prefix; set of discovered paths (provenance)            |
> | $\pi_B,\pi_T,\pi_S$; $a_t^B,a_t^T,a_t^S$                     | Policies and actions for Edit/Traverse/Curate                |
> | $s_t=(q,G_t,\mathcal{F}_t,\mathcal{P}_t,\mathbf{b}_t)$       | State summary at step $t$; $\mathbf{b}_t$: remaining budgets |
> | $\beta=(\beta_{\text{edge}},\beta_{\text{lat}},\beta_{\text{tok}})$ | Episode budgets: edges, latency (steps), selected tokens     |
> | $C_{\text{edge}},C_{\text{lat}},C_{\text{tok}}$; $c_t^{(k)}$ | Cumulative costs; per‑step increments ($\sum_t c_t^{(k)}=C_k$) |
> | $\boldsymbol{\lambda}=(\lambda_{\text{edge}},\lambda_{\text{lat}},\lambda_{\text{tok}})$ | Lagrange multipliers (resource prices)                       |
> | $R_{\mathrm{acc}}(\tau)$; $r^{\mathrm{acc}}_t$; $r'_t$       | Episode reward; per‑step reward; shaped return $r'_t$ (see Appendix B) |
> | $\mathcal{L}(\pi,\boldsymbol{\lambda})$                      | Lagrangian $\mathbb{E}[R_{\mathrm{acc}}-\boldsymbol{\lambda}^{\top}C]$ |
> | $Q^{\text{task}},Q^{\text{edge}},Q^{\text{lat}},Q^{\text{tok}}$ | Centralized‑critic action‑values (task and cost heads)       |
> | $A^{i,h}_t,\ A^{i,\lambda}_t$; $i\in\{B,T,S\}$               | Counterfactual and Lagrangian‑shaped advantages; agent index |
> | $D^\star$; $\operatorname{tok}(\cdot)$                       | Ordered curated evidence; token‑count operator               |
> | $e=(u,r,v)$                                                  | KG triple (head $u$, relation $r$, tail $v$)                 |
> |$H,\ \bar d,\ \lvert \mathcal{P}_t \rvert,\ K$  |  Hop cap; average local branching factor; candidate pool size; selected list length ($K$ dynamic)                                              |
>
> ---
>
> > **Q2.** The experiment part is solely conducted with Qwen3-32B, I wonder whether the author has tried as LLM?
>
> **A2.** CLAUSE is reader-agnostic: the three agents only construct and curate the context, and any decoder can consume the selected snippets under the same token budget. In the submission we fixed the reader to Qwen3-32B to isolate gains from context construction; we already include no-retrieval baselines for GPT-OSS-120B and LLaMA3.3-70B (Table 2). In the revision, we have added a reader-swap sensitivity analysis in Appendix D (Tables 8 and 9) by plugging LLaMA- and GPT-class readers into the exact same pipeline and budgets, and we observe that CLAUSE's relative gains persist with different readers. No CLAUSE component or prompt is model-specific; the agent encoders are lightweight MLPs over symbolic features and do not call the reader during construction.
>
> **Table II: EM@1 of CLAUSE with different reader LLM backbones on all benchmarks.**
>
> ---
>
> | Reader backbone | HotpotQA | FactKG | MetaQA 1-hop | MetaQA 2-hop | MetaQA 3-hop |
> | --------------- | -------- | ------ | ------------ | ------------ | ------------ |
> | Qwen3-32B       | 71.7     | 84.2   | 91.0         | 87.3         | 85.5         |
> | LLaMA3.3-70B    | 73.5     | 85.0   | 92.7         | 87.7         | 87.2         |
> | GPT-OSS-120B    | 75.0     | 85.8   | 92.2         | 88.0         | 87.5         |
>
> ---

---

> > ### Comment · Reviewer_2uqG · 2025-11-23
> >
> > Thanks, I appreciate the effort the author made, I will keep my score as it is already positive

---

### Author Response · Authors · 2025-12-02
**General response to the Area Chair**

We would like to thank the Area Chair for overseeing the review process and for the careful evaluation of both our manuscript and our response to the reviewers’ comments. We would also like to thank all reviewers for their thoughtful assessments and fast responses. Our initial scores were (6, 8, 6, 2) (avg 5.5). **After our rebuttal and revisions, Reviewer Aqnr raised the score from 2 to 6, leading to (6, 8, 6, 6) (avg 6.5)**.

---

### TL;DR

> **CLAUSE** is a three‑agent neuro‑symbolic framework for multi‑hop KGQA that learns how to build context (which edges to add/delete, which paths to follow/backtrack, which snippets to keep, and when to stop) under explicit per‑query budgets on edges, interaction steps, and tokens. It uses a constrained multi‑agent PPO variant (**LC‑MAPPO**) with a centralized critic and dual variables so a single policy can trade off accuracy vs. latency/token cost at test time.

---

### Contributions

1. **[Core Idea]** Multi-hop KGQA is cast as requirements-conditioned context assembly under explicit per-query budgets on edits, steps, and selected tokens, enabling controllable accuracy–efficiency trade-offs.
2. **[Novel Framework]** We propose CLAUSE, an agentic neuro-symbolic controller with symbolic, auditable actions guided by lightweight neural scoring for subgraph editing, navigation/backtracking, evidence retention, and stopping.
3. **[Training]** We adapt constrained reinforcement learning through LC-MAPPO with centralized training and decentralized execution, a multi-head critic separating task and cost values, and dual variables to enforce or trade off budgets.
4. **[Strong Performance]** Experiments on various KGQA datasets show the state-of-the-art performance of CLAUSE.

---

### Main Response

**[For reviewer 2uqG]** – rating: 6 (kept positive after rebuttal)

> reviewer 2uqG replied: "Thanks, I appreciate the effort the author made, I will keep my score as it is already positive”

* Moved symbol table from appendix into main text.
* Explicitly defined all key symbols.
* Added experiments with LLaMA3.3‑70B and GPT‑OSS‑120B showing CLAUSE’s relative gains persist across readers.

**[For reviewer L2iz]** – rating: 8 (no score change)

* Distilled contributions into four points (formulation, framework, training, evidence).
* Expanded Method section to clarify the three agents and their interaction.
* Added explicit comparison of LC‑MAPPO to existing constrained/Lagrangian RL.

**[For reviewer aDuz]** – rating: 6 (no score change)

* Clarified that inference uses a single bounded reader LLM call per query.
* Explained lightweight controllers scale with per‑query budgets, not KG size.
* Described LC‑MAPPO training: small actors + centralized critic, frozen reader, offline trajectories.
* Added KG construction/scale details and scalability analysis.
* Described failure‑mode detection and budget‑safe recovery strategies.

**[For reviewer Aqnr]** – rating: 2 → 6 (after rebuttal)

> reviewer Aqnr replied: "Thank you for the comprehensive response, and for the aditions to the paper.  ....  I will update my score accordingly."

* Clarified problem formulation (KG, question‑conditioned and frontierized subgraphs) and cleaned notation.
* Consolidated convergence assumptions into an appendix and explained how they are satisfied.
* Detailed roles and architectures of the three agents.
* Removed code‑level function names and replaced with architecture‑level descriptions.

---

### Meta-Review · Area_Chair_zbHb · 2025-12-31

**Summary:**

In this paper, the authors propose CLAUSE, an agentic neuro-symbolic framework for multi-hop question answering over knowledge graphs. The key idea is to treat context construction as a sequential decision process under explicit per-query budgets on graph expansion, interaction steps, and token usage. The framework uses three coordinated agents to control subgraph construction, path navigation, and evidence selection, trained with a constrained multi-agent reinforcement learning approach. Overall, reviewers found the problem important and the approach novel, and the rebuttal addressed most major concerns.

**Reviewer Concerns:**

Most reviewer concerns were addressed in the rebuttal and revision.

Several reviewers initially raised issues about clarity and exposition, including unclear notation, missing definitions (e.g., KG, frontier, agents, state), and insufficient description of model components. These points were addressed by adding a notation table, clarifying the problem formulation, and providing detailed descriptions of the three agents, their architectures, and training procedure.

There were also concerns about the novelty and necessity of LC-MAPPO, as well as whether existing constrained RL methods could be used instead. The authors clarified that LC-MAPPO is an adaptation rather than a new theoretical algorithm, and provided ablation results and discussion showing limitations of simpler constrained RL baselines in this multi-agent, multi-budget setting.

There were also questions regarding training cost, scalability, and behavior on large or noisy knowledge graphs. They were addressed by adding complexity analysis, training cost discussion, and additional explanations in the appendix. The authors also clarified that inference uses a single bounded reader call per query.

One reviewer initially questioned the soundness and presentation and gave a low score. After the rebuttal and added clarifications, this reviewer updated their score significantly, indicating that the main technical concerns were resolved.

Some minor concerns remain, such as system complexity and limited evaluation on very large real-world KGs, but these are not critical and do not block acceptance.

**Reviewer Scores:**

Reviewer 2uqG: Initially positive (6) and indicated that the rebuttal addressed the raised concerns. The score would likely remain 6.

Reviewer L2iz: Strongly positive (8) with no major concerns after rebuttal. The score would remain 8.

Reviewer aDuz: Generally positive (6), with minor concerns about complexity and scalability that were addressed in the rebuttal. The score would likely remain 6.

Reviewer Aqnr: Initially negative (2) due to clarity and formulation issues, but after the rebuttal and revisions, the reviewer explicitly mentioned that they would raise their score. The authors claimed that the reviewer has already updated their score to 6 before the leakage. So, likely the reviewer would have raised their score to 6 after the rebuttal.

---

### Decision · Program_Chairs · 2026-01-26

Accept (Poster)